# Biosynthetic production of anticoagulant heparin polysaccharides through metabolic and sulfotransferases engineering strategies

Jian-Qun Deng[1], Yi Li[1], Yu-Jia Wang[1], Ya-Lin Cao[1], Si-Yu Xin[1], Xin-Yu Li[1], Rui-Min Xi[1], Feng-Shan Wang[1,2] & Ju-Zheng Sheng [1,2] ✉

Heparin is an important anticoagulant drug, and microbial heparin biosynthesis is a potential alternative to animal-derived heparin production. However, effectively using heparin synthesis enzymes faces challenges, especially with microbial recombinant expression of active heparan sulfate *N*-deacetylase/*N*-sulfotransferase. Here, we introduce the monosaccharide *N*-trifluoroacetylglucosamine into *Escherichia coli* K5 to facilitate sulfation modification. The Protein Repair One-Stop Service-Focused Rational Iterative Site-specific Mutagenesis (PROSS-FRISM) platform is used to enhance sulfotransferase efficiency, resulting in the engineered NST-M8 enzyme with significantly improved stability (11.32-fold) and activity (2.53-fold) compared to the wild-type *N*-sulfotransferase. This approach can be applied to engineering various sulfotransferases. The multienzyme cascade reaction enables the production of active heparin from bioengineered heparosan, demonstrating anti-FXa (246.09 IU/mg) and anti-FIIa (48.62 IU/mg) activities. This study offers insights into overcoming challenges in heparin synthesis and modification, paving the way for the future development of animal-free heparins using a cellular system-based semisynthetic strategy.

The chemoenzymatic synthesis of heparin, leveraging heparin biosynthesis enzymes, presents a promising avenue for addressing vulnerabilities in the supply chain of animal-derived therapeutic heparin[1,2]. Heparin, a natural anticoagulant, holds pivotal importance in preventing and treating blood clots across various medical scenarios including deep vein thrombosis, pulmonary embolism, and surgical interventions. The *N*-sulfation modification of heparin has critical significance for subsequent modifications[3], which dictate the physiological and pharmacological activities of heparin[4]. The presence of the glucosamine sulfate (GlcNS) residue is indispensable for all subsequent modifications during heparin biosynthesis[5]. C5-epimerases and all the necessary heparin sulfotransferases, excluding the *N*-deacetylase domain of the bifunctional heparan sulfate *N*-deacetylase/*N*-sulfotransferase (NDST), have been expressed and exhibited activity in *Escherichia coli*[6]. Recently, high-resolution cryogenic-electron microscopy (cryo-EM) structures of NDST1 have been obtained[7,8]; the rich structural information will enhance engineering applications of NDST. However, the expression of active deacetylase proteins remains unattainable in the cost-effective *E. coli* expression system, and even in yeast[9] and insect cell/baculovirus expression systems[10].

To address the deficiency in *N*-deacetylase activity, the present in vitro chemoenzymatic approach employed uridine diphosphate-*N*-trifluoroacetylglucosamine (UDP-GlcNTFA) instead of the natural substrate of α−1,4-*N*-acetylglucosaminyltransferase (KifA)[11,12] or *Pasteurella multocida* heparosan synthase 2 (PmHS2)[13], uridine diphosphate-*N*-acetylglucosamine (UDP-GlcNAc). In this method, UDP-GlcNTFA undergoes polymerization with uridine diphosphate-ᴅ-glucuronic acid (UDP-GlcA), and then the trifluoroacetyl group can be efficiently removed by mild alkaline treatment[14]. *N*-sulfation was achieved by incorporating a *Homo sapiens N*-sulfotransferase (NST)

---

[1]School of Pharmaceutical Sciences, Shandong University, Jinan, China. [2]National Glycoengineering Research Center, Shandong University, Jinan, China. ✉e-mail: shengjuzheng@sdu.edu.cn

domain with substantial activity. The oligosaccharides containing GlcNS are isomerized at the C5 position[5], followed by the application of three sulfotransferases (2-OST[14], 6-OST[15], and 3-OST[16]) and a sulfonate donor (3′-phosphoadenosine-5′-phosphosulfate, PAPS) to achieve sulfation modification at the O position of the sugar chain, thus synthesizing structurally uniform anticoagulant heparin oligosaccharides (Supplementary Fig. 1). This chemoenzymatic synthesis strategy offers a safe and more economical alternative to the conventional animal-derived heparin supply chain. By using alternative substrates to overcome the challenges associated with expressing functional NDST, this approach facilitated large-scale synthesis of heparin that retains the essential structural and pharmacological properties[17].

The bacterial capsular polysaccharide of *E. coli* strain K5, referred to as heparosan, comprises a repeating −4-GlcA-$\beta$−1,4-GlcNAc-$\alpha$−1- structure, identical to the nonsulfated precursor of heparin[18]. This precursor, recognized as a readily available starting material for heparin synthesis[19,20]. An artificial biosynthetic pathway for UDP-GlcNAc, identified in vitro, has been found to have broad tolerance for substrate modification, with the C2′ derivatives of GlcNAc serving as tolerant substrates for this salvage pathway. In previous studies, we disrupted the UDP-GlcNAc and heparosan synthesis pathways in heparosan-synthesizing bacteria and introduced the exogenous heparin synthase PmHS2 along with a UDP-GlcNAc salvage synthesis pathway via *N*-acetylhexosamine 1-kinase (NahK)[21] and UDP-GlcNAc pyrophosphorylase 1 (AGX1)[22]. Through this approach, we replaced the conserved GlcNAc residue in heparosan on the surface of *E. coli* K5 with the analog *N*-azidoacetylglucosamine (GlcNAz). Here, to overcome the issue of inactive NDST, we applied the same metabolic strategy to introduce the non-natural monosaccharide *N*-trifluoroacetylglucosamine (GlcNTFA) into the capsule of engineered *E. coli* K5. Subsequently, the extracted heparosan polysaccharide underwent mild, weak-base treatment to expose the amino groups, rendering it suitable for the necessary sulfation modifications essential for heparin synthesis.

The next challenge that must be overcome in microbial heparin synthesis is unsatisfactory stability and activity of heparin sulfotransferases. Undoubtedly, the advancement of effective heparin sulfotransferases is a key driving force in the development of glycosaminoglycan (GAG) drugs—researching and discovering more efficient heparin sulfotransferases will help to enhance the efficacy and safety of these drugs, ultimately promoting their development[23]. The highly regioselective and stereoselective properties of these enzymes make them exceptional and captivating catalysts[24]. Capitalizing on their catalytic capacity is an appealing prospect for sustainable applications[25], facilitating the synthesis of tailor-made GAG oligosaccharides for comprehensive pharmacological and mechanistic studies[26]. However, the instability of most heparin sulfotransferases of eukaryotic origin when expressed in prokaryotic systems, coupled with unintended effects leading to enzyme misfolding and protein aggregation, is a significant challenge[27]. Consequently, robust enzymes are highly desirable for industrial needs, but require engineering to overcome instability in convenient prokaryotic expression systems.

In recent years, the rapid advancement of various algorithms has enhanced our ability to manipulate the structure and function of biological macromolecules[28]. Building on the success of designing individual stable mutations in enzymes, several iterative strategies have emerged to produce robust mutants that effectively integrate information from various evolutionary or energetically function-based approaches[29]. While liquid chromatography methods have been developed to screen sulfotransferase activity[30], this screening step remains labor-intensive, and conventional mutant libraries are often too extensive for practical screening. The primary challenge in the directed evolution of sulfotransferases lies in generating high-quality mutant libraries, achievable through reliable computational methods, that necessitate minimal screening of enzymes in the laboratory[31]. In

this context, leveraging a eukaryotic NST as a paradigm, we endeavor to design methodology of PROSS-FRISM by seamlessly integrating high-throughput activity screening, computational engineering, and a streamlined codon mutant library. Our overarching goal remains steadfast: to engineer a swift, practical, and resource-efficient solution that not only yields mutant enzymes with superior properties but also pushes the boundaries of experimental efficiency.

Here, we synthesize *N*-sulfated heparin polysaccharides as the foundational material for anticoagulant heparin biosynthesis. We achieve this using metabolically engineered *E. coli* K5 coupled with protein-engineered NST (Fig. 1). This methodology avoids the use of NDST, streamlines the production process and reduces production costs. Overall, it enables the direct generation of heparins with potent anticoagulant properties. The optimized synthesis process not only enhances economic efficiency but also ensures high-quality and sustainable operations.

## Results
### Metabolic preparation of *N*-sulfated heparosan polysaccharides using engineered *E. coli*

In previous investigations, we engineered *E. coli* K5 strains (Supplementary Table 1) capable of producing heparosan polysaccharides with non-natural structures[32,33]. This achievement involved disrupting the de novo biosynthesis of UDP-GlcNAc in *E. coli* and introducing a constitutively expressed UDP-GlcNAc synthesis pathway along with an inducible exogenous heparin synthase (PmHS2)[13]. GlcNTFA, akin to GlcNAz, is a GlcNAc analog and is commonly substituted for GlcNAc in heparin in vitro chemoenzymatic synthesis. It facilitates the deacylation of *N*-acetyl groups to replace the function of *N*-deacetylase (NDase)[34]. GlcNTFA is tolerated by the UDP-GlcNAc biosynthetic machinery comprising NahK from *Bifidobacterium longum* and *Homo sapiens* AGX1, as well as by the heparin synthase PmHS2 (Fig. 2A). We anticipated that polysaccharide with -4-GlcA-$\beta$−1,4-GlcNTFA-$\alpha$-1- structure, generated by incorporating GlcNTFA into the heparosan capsule, could be easily degraded to -4-GlcA-$\beta$−1,4-GlcNH$_2$-$\alpha$-1- structure through removal of the trifluoroacetyl (TFA) group by using a weak base. The resulting polysaccharide can then be further modified by sulfotransferase to produce active heparin.

To minimize the GlcNAc to GlcNTFA ratio within the capsule, *E. coli* K5 was initially cultivated in a medium supplemented with GlcNAc. Subsequently, washed cells were transferred to a medium containing 1,3,4,6-tetra-*O*-acetyl-2-deoxy-2-trifluoroacetylamino-D-glucopyranose (Ac$_4$GlcNTFA)[35], and isopropyl $\beta$-D-thiogalactopyranoside (IPTG) to induce the expression of NahK, AGX1, and PmHS2 for polysaccharide fermentation. From Luria-Bertani (LB) culture medium supplemented with Ac$_4$GlcNTFA, we extracted 30 mg/L of polysaccharides. Because GlcNAc and GlcNTFA have similar properties and the same retention time (~25.5 min) on a polyamine column, while GlcNAc cannot be deacetylated by LiOH, to determine the proportion of GlcNTFA incorporated into the polysaccharides, we treated them with LiOH to remove TFA from GlcNTFA and form GlcNH$_2$ (Fig. 2B). Analysis of disaccharides by polyamine-based anion exchange-high-performance liquid chromatography (PAMN-HPLC) was conducted after complete degradation of the polysaccharide using heparin endonuclease heparinase III[36] (HepIII). HepIII specifically targets the $\alpha$-1,4 linkage between GlcN and GlcA residues. The analysis revealed that approximately 74% of the composition was -GlcA-GlcNH$_2$-, indicating that GlcNTFA had replaced around 74% of the GlcNAc in the polysaccharide (Fig. 2C). Upon addition of the sulfate donor PAPS and recombinant NST, GlcNH$_2$ on the polysaccharide was completely converted to GlcNS, as confirmed by the liquid-phase analysis of disaccharides obtained after degradation by Hep III, where 4,5 unsaturated uronic acid ($\Delta$UA)-GlcNH$_2$ was completely converted to UA-GlcNS. The resulting disaccharides were examined by high-performance liquid chromatography-mass spectrometry (HPLC-MS)

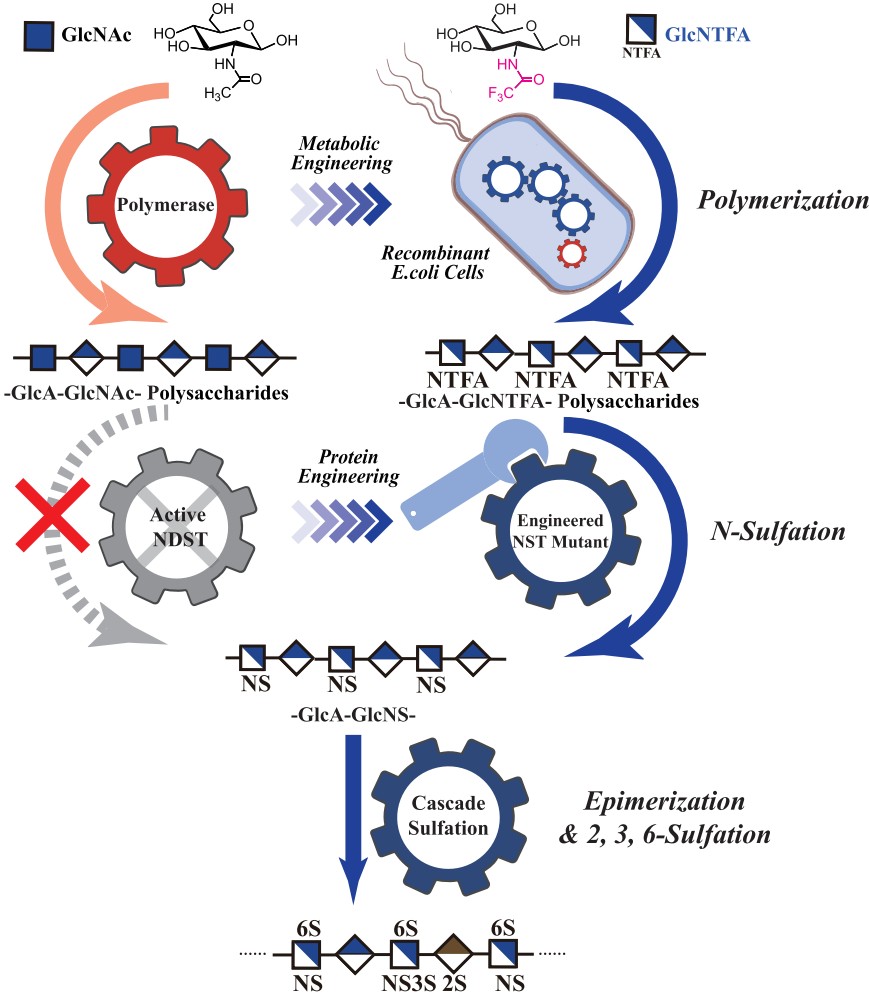

**Fig. 1 | Heparin synthesis employing an in vitro/in vivo combination strategy.**
The left diagram illustrates the conventional method for -GlcA-GlcNS- heparin preparation. *N*-acetylglucosamine (GlcNAc) undergoes polymerization with glucuronic acid (GlcA), catalyzed by heparin backbone synthase. Subsequently, the precursor polysaccharide -GlcA-GlcNS- undergoes deacetylation and *N*-sulfation under the influence of *N*-deacetylase/*N*-sulfotransferase (NDST). Expression of active recombinant NDST has historically been a significant bottleneck in achieving this process in prokaryotic expression systems. The right diagram represents the method proposed in this study. GlcNAc analog *N*-trifluoroacetylglucosamine (GlcNTFA) is taken up by the *E. coli* K5ASSH strain and utilized for the production of -GlcA-GlcNTFA- polysaccharides, which are then converted to -GlcA-GlcNS- polysaccharides through chemical enzymatic synthesis using an engineered highly stable *N*-sulfotransferase (NST) mutant (NST-M8). -GlcA-GlcNS- polysaccharides are ultimately modified into anticoagulant polysaccharides by a cascade sulfation system.

(Supplementary Fig. 2). The structures of the digestion products were identified as disaccharides with molecular masses of 337.21 ± 0.2 Da (ΔUA-GlcNH$_2$; Fig. 2D), 378.94 ± 0.2 Da (ΔUA-GlcNAc; Fig. 2E), and 417.07 ± 0.2 Da (ΔUA-GlcNS; Fig. 2F; calculated molecular masses: ΔUA-GlcNH$_2$, 337.10 Da; ΔUA-GlcNAc, 379.11 Da; and ΔUA-GlcNS, 417.06 Da). This result confirms the integration of the GlcNAc analog GlcNTFA into the heparosan polysaccharide. The polysaccharide -GlcA-GlcNH$_2$-, with the trifluoroacetyl group removed, can be converted to -GlcA-GlcNS- heparin polysaccharide by NST. This scalable strategy has the potential for production of active heparins of non-animal origin.

## PROSS-FRISM engineering *N*-sulfotransferase

The -GlcA-GlcNS- structure is imperative for bioengineered heparin to be active. However, microbial expression of the bifunctional NDST, responsible for catalyzing the first modification reaction, has consistently been a significant bottleneck[37]. To overcome this limitation, we have metabolically introduced GlcNTFA into *E. coli* capsular polysaccharides using an engineered strain, K5ASSH. This approach facilitates the straightforward preparation of deacetylated heparin oligosaccharides through fermentation. NST, the initial sulfotransferase for heparin modification, is the C-terminal domain of *Homo sapiens* NDST (residues L557–R882)[38]. Although NST can be recombinantly expressed in an active form in prokaryotic systems, its limited stability restricts its scalable application[39]. Although the *E. coli* expression system is presently the most economical and convenient for protein expression, the enzymatic activity of many GAG sulfotransferases expressed in *E. coli*, including NST, is much lower than of that expressed in yeast and insect cells[9]. Consequently, here, we aimed to establish an evolutionary platform, exemplified by NST, that can swiftly and efficiently accumulate beneficial variants from a specific eukaryotic source of sulfotransferase in a well-defined library that enables stable prokaryotic expression. A further complication is that the sulfotransferase reaction lacks a fluorescence assay, which prompted the development of a liquid chromatography method using a chemoenzymatically synthesized pentasaccharide as a receptor substrate. However, the current analysis rate of 30 samples per day is insufficient for high-throughput screening of sulfotransferase

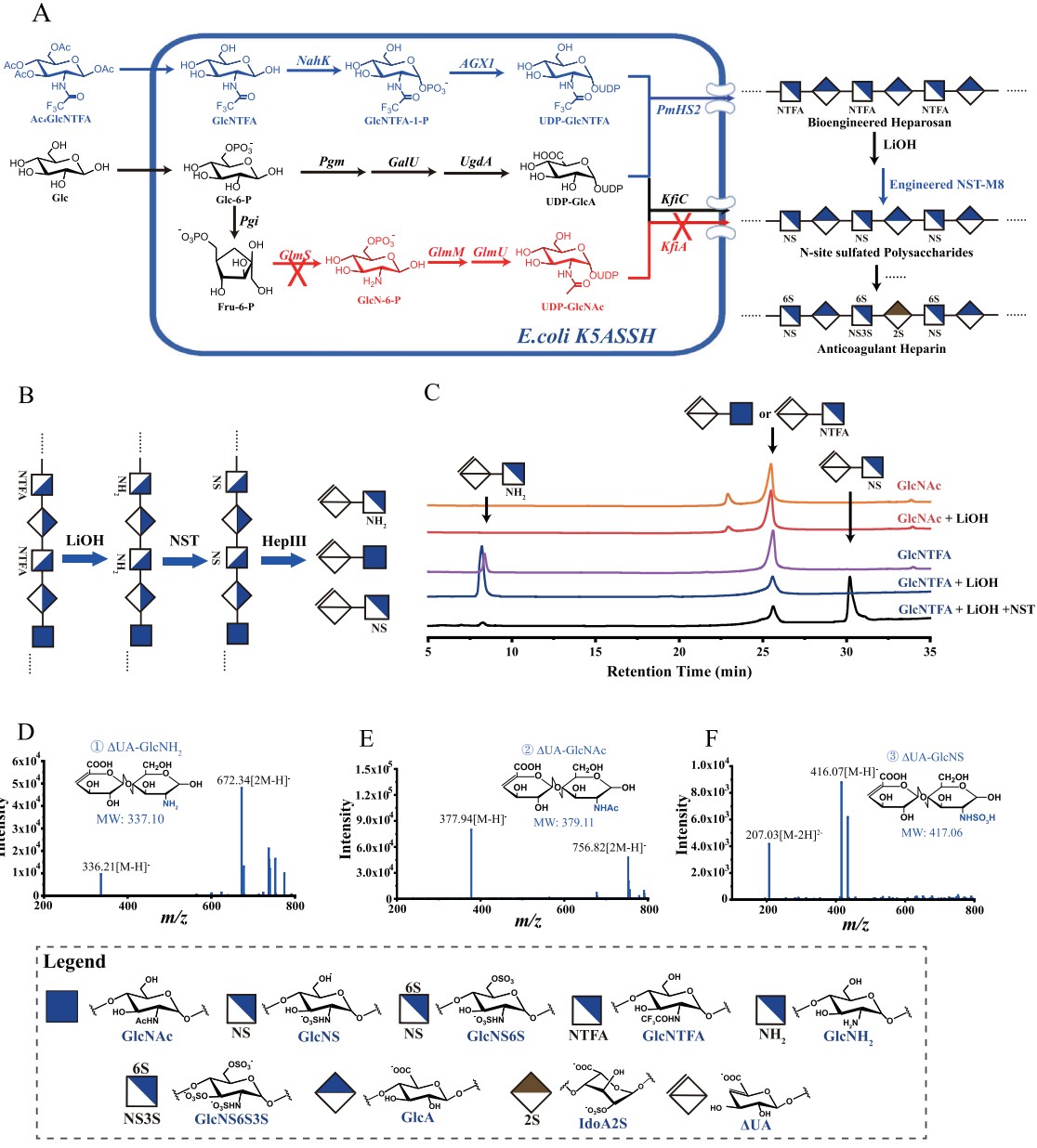

**Fig. 2 | Integrated strategy for *N*-sulfated polysaccharide synthesis and analysis. A** By disrupting the *E. coli* kfiA gene to impede heparosan polysaccharide biosynthesis and the glmS gene to halt the UDP-GlcNAc biosynthesis pathway, the non-natural sugar GlcNTFA is incorporated into capsular polysaccharide via the rescue pathway catalyzed by NahK and AGX1. These enzymes convert GlcNTFA into an activated nucleotide sugar, which is polymerized into a -GlcNTFA-GlcA- structured polysaccharide by PmHS2. PmHS2 *Pasteurella multocida* heparosan synthase 2, KfiA UDP-*N*-acetyl-ᴅ-glucosamine:heparosan α-1,4-*N*-acetyl-ᴅ-glucosaminyltransferase, KfiC UDP-glucuronic acid: heparosan β-1,4-glucuronosyltransferase, GlmS glutamine-fructose-6-phosphate transaminase, Pgm phosphoglucomutase, GalU UTP-glucose-1-phosphate uridylyltransferase, UgdA UDP-glucose 6-dehydrogenase, GlmM phosphoglucosamine mutase, GlmU bifunctional GlcNAc-1-phosphate uridyltransferase/glucosamine-1-phosphate acetyltransferase, NahK *N*-acetylhexosamine 1-kinase, AGX1 UDP-GlcNAc pyrophosphorylase 1, NST *N*-sulfotransferase. **B** Schematic illustration depicting the synthesis of *N*-sulfated heparin polysaccharides through an in vitro/in vivo combination strategy. **C** Analysis of disaccharides resulting from LiOH treatment and degradation of bioengineered

heparosan (catalyzed by HepIII) using polyamine-based anion exchange (PAMN)-high-performance liquid chromatography (HPLC) with detection at 232 nm. Retention times: ΔUA-GlcNH₂, ~7.0 min; ΔUA-GlcNAc, ~25.5 min; ΔUA-GlcNS, ~30.5 min. Orange: Polysaccharides from *E. coli* K5 ASSH fed GlcNAc. Red: Polysaccharides from *E. coli* K5 ASSH fed GlcNAc; the extracted polysaccharides were treated with LiOH. Purple: Polysaccharides from *E. coli* K5 ASSH fed GlcNTFA. Blue: Polysaccharides from *E. coli* K5 ASSH fed GlcNTFA; the extracted polysaccharides were treated with LiOH. Black: Polysaccharides from *E. coli* K5 ASSH fed GlcNTFA; after treating the extracted polysaccharides with LiOH they were reacted with 3-phosphonoadenosine-5-phosphosulfate (PAPS) and NST. Polysaccharides treated with NST and PAPS showed the disappearance of the ΔUA-GlcNH₂ chromatographic peak and the emergence of a ΔUA-GlcNS peak. **D**–**F** Electrospray ionization-mass spectra of disaccharides produced by degradation of bioengineering heparosan with HepIII. **D** ΔUA-GlcNH₂ (*m/z* 336.05 ± 0.2 Da [M−H]⁻). **E** ΔUA-GlcNAc (*m/z* 378.10 ± 0.2 Da [M−H]⁻). **F** ΔUA-GlcNS (*m/z* 416.05 ± 0.2 Da [M−H]⁻). Source data are provided as a Source Data file.

mutants. To address this, a PAPS regeneration system[40] using recombinant rat liver aryl sulfotransferase IV (ASTIV) has been constructed to convert 3-phosphonoadenosine-5-phosphate (PAP) to PAPS, with *p*-nitrophenyl sulfate (*p*NPS) serving as the sulfate donor. The

quantification of sulfotransferase activity can be achieved by measuring the change in absorption kinetics of *p*NP, at 405 nm[41].

We endeavored to establish a high-throughput screening method for sulfotransferases and assessed its efficacy by incorporating PAPS,

*p*NPS, ASTIV, deacetylated heparin pentasaccharide and IPTG-induced NST expression fragmentation supernatant (without IPTG induction in the control group) in microtiter plates. The sulfotransferase activity was evaluated by monitoring the kinetic changes in 405 nm absorption (*p*NP generation). Unfortunately, the calculated z-factor was only −1.83, signifying inadequate screening capability because of the low activity of ASTIV, which was a bottleneck for practical application. Using PROSS[42], a mutant enzyme design strategy that integrates force-field-based Rosetta modeling with phylogenetic sequence information, we engineered ASTIV(M), which had 52 mutations compared with the wild-type (WT) enzyme. This design led to higher expression levels (40 mg/L; Supplementary Fig. 3), increased thermal stability, and enhanced catalytic efficiency. The 3-fold rise in ASTIV activity substantially elevated the signal value of the positive control in the sulfotransferase assay, resulting in a z-factor of 0.77 (Supplementary Fig. 4). This enhancement makes the assay suitable for high-throughput screening of sulfotransferases.

The expression of eukaryotic sulfotransferases in prokaryotic systems presents significant challenges. First, the absence of glycosylation modification in proteins recombinantly expressed in prokaryotic systems necessitates compensatory mutations to maintain protein stability[43]. Second, the membrane localization of the Golgi apparatus results in the proteins being situated on a large hydrophobic surface in eukaryotes, posing difficulties for sulfotransferase expression in *E. coli* and leading to issues of activity instability[44]. Third, a delicate balance between activity and stability is required. The overall stability of the enzyme is governed by an intricate network of favorable intramolecular interactions, including externally exposed hydrophilic residues, stable stacking of the internal hydrophobic core, and a network of hydrogen bonds between secondary structures[45]. However, mutations in active-site residues, essential for ligand binding and catalysis, may jeopardize stability by disrupting the network of intramolecular interactions that collectively control protein stability[46,47]. Recombinant NST proteins exhibit high catalytic efficiency but poor stability, with a half-life of <0.5 days. It is crucial that, in mutagenesis of the enzyme, enhancing stability should not compromise functional activity. Given this context, our focus to improve the stability of NST without compromising its catalytic activity centered on addressing the adverse effects of surface hydrophobic residues and the instability of surface secondary structures (Fig. 3A).

The library generated by irrational design yields a large and highly random number of mutants. Despite our efforts to optimize screening methods, the process remains labor-intensive. Therefore, we prefer employing rational design methods to narrow down the range of amino acid mutations, resulting in a size-reduced mutation library. In this study, we used computational and experimental approaches to develop the NST protein with high thermal stability as a template and subsequently designed the PROSS-FRISM evolutionary strategy. Through structural analysis and molecular dynamics simulations, candidate mutation sites were identified based on stability and sequence conservation, thus establishing the subsequent design scope (total 39 hot sites). The PROSS engineering approach facilitated the efficient evolution of computationally stable mutations, while the FRISM strategy effectively guided iterative mutations of residual hotspot sites. By combining computational prediction, experimental validation, and iterative optimization, we designed NST mutants with enhanced stability. Therefore, the PROSS-FRISM strategy holds promise as a universal directed evolution tool for eukaryotic proteins expressed in prokaryotic systems.

We aim to establish a hotspot amino acid library based on the impact of glycosylation deficiency in NST expressed by *E. coli* and sequence conservation analysis. The structure of the human NST protein (PDB accession 1NST)[48] expressed in *E. coli* has been resolved, but information about its glycosylation is lacking. NST has only one *N*-glycosylation site (residue N667), and we attempted to model the glycosylated NST structure. The structure of C5-epi, a eukaryotic protein with C5-epimerase activity, has been resolved (PDB accession 6HZZ)[49], and the polysaccharide at position N393 provides a detailed structural example of *N*-glycosylated structure. Here, we manually grafted the polysaccharide from residue N393 of C5-epi onto residue N667 of NST and hence obtained structural models of NST with and without glycosylation. Molecular dynamics (MD) simulations of 20 ns were performed on glycosylated and non-glycosylated NST using Amber (Supplementary Fig. 5). In identifying candidate amino acid residues for mutagenesis, to ensure reliability while covering a comprehensive array of residues, a Δroot-mean-square fluctuation value of <0.6 in the MD simulations before and after glycosylation, along with a score of <8 for the residue in sequence conservation analysis (values range from 0 to 10, with 10 indicating complete amino acid conservation at that position) were applied as thresholds (Supplementary Fig. 6). Application of these criteria yielded a total of 39 unique predicted surface mutation sites as potentially stable mutation candidates in NST (Supplementary Fig. 7).

PROSS engineering achieves the maximum accumulation of mutations. However, iterative saturated evolution of up to 39 sites would require a substantial physical effort, limiting the scope for creating stable biocatalysts through stepwise combinations alone. Hence, knowledge-guided approaches are essential to navigate the combinatorial sequence space. PROSS engineering facilitates the selection of computationally stable mutations and the rapid implementation of optimized combinations of combinatorial sequences within the space of these mutations. This approach can address a wide range of stability deficiencies in various proteins without compromising their molecular activity. Focusing on the 39 sites identified above, we selected four mutants, NST-designs 1–4 (with 6, 11, 15, and 22 mutation sites, respectively) for experimental validation (Supplementary Figs. 8 and 9 and Supplementary Table 2). Among them, NST-design 3 (NST-M1, having 15 mutations compared with the WT enzyme) exhibited the highest expression level and stability, and served as the evolutionary starting point for the next generation (Fig. 3B).

To iterate the remaining 24 hot sites onto the NST-M1 template as efficiently as possible, we are incorporating computer-assisted methods and adopt the Focused Rational Iterative Site-Specific Mutagenesis (FRISM) strategy to make the selection of mutation sites and the generation of mutation libraries more rational and efficient. Protein design of potentially stable further point mutations was then performed using Rosetta_ddg[50]. The ΔΔG values of single-point mutations at the remaining 24 of the 39 candidate mutagenesis sites (see above) in NST-M1 were calculated by individually substituting them with all proteinogenic amino acid residues (Fig. 3C and Supplementary Table 3). Among them, T669, D682, V685, A691, H769, A770, Q772, H805, and G823 did not have mutations with lowered ΔΔG values. The remaining 15 sites were designed with streamlined codon libraries for stability screening according to the reduction in ΔΔG values. This rational design process resulted in not only a substantial decrease in the amount of screening required but also a significant increase in the probability of obtaining beneficial mutants. Using the established high-throughput activity assay based on ASTIV, the top 5% of mutants showing significant kinetic changes in 410 nm absorption were sequenced and revalidated for activity. A total of 17 single-point mutants were identified, which exhibited enhanced stability compared to NSTM1. However, due to the possibility of mutually interfering effects of mutations in adjacent amino acids, positive mutations may fail to cooperate and achieve the desired functionality. To address this issue, the 17 beneficial mutations were grouped into four clusters based on their close proximity and potential interactions such as hydrogen bonding and salt bridges (Supplementary Fig. 10). Each

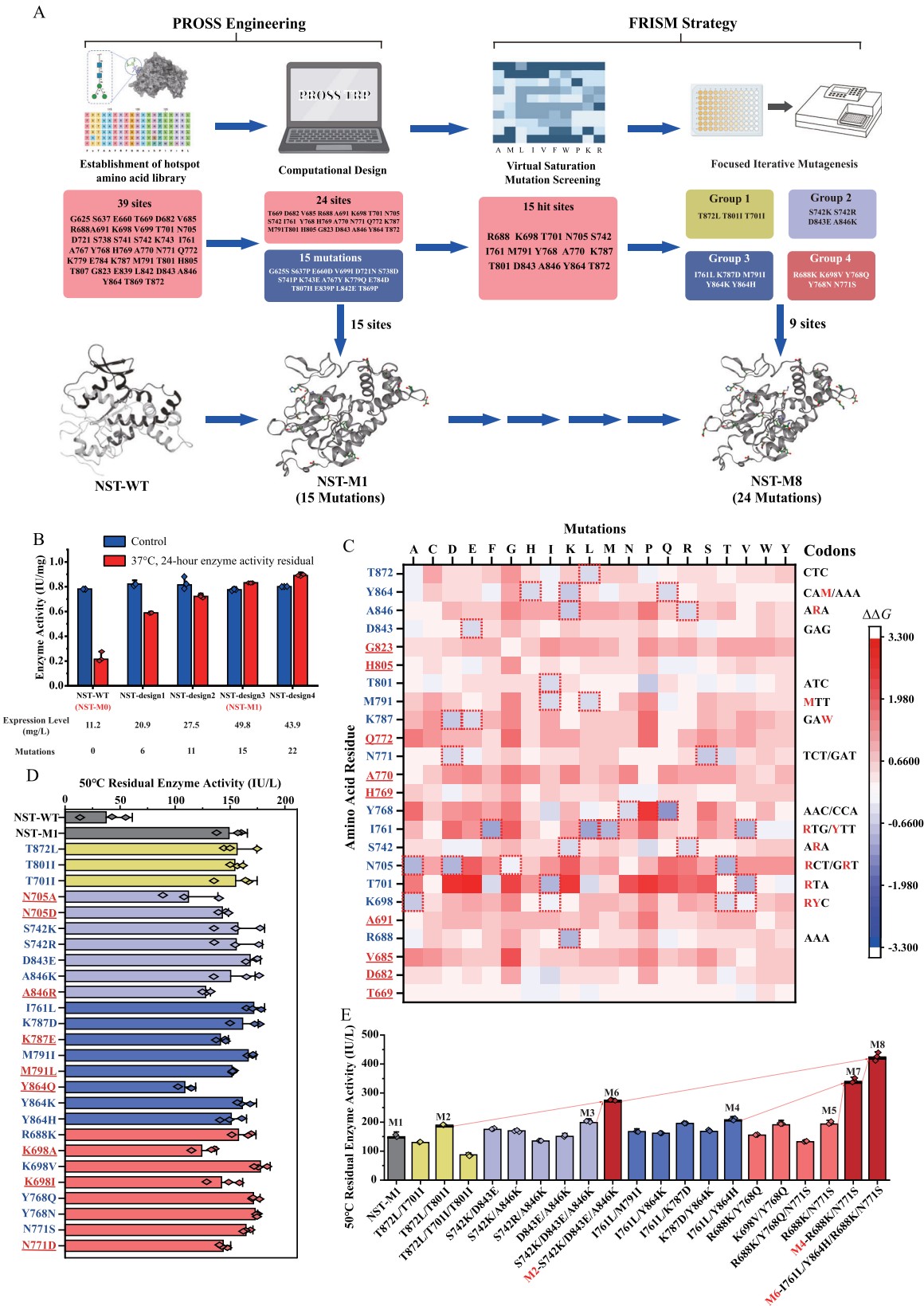

cluster underwent screening with a gene recombination constructed mutant library[51] containing permutations of the necessary single-point mutations (Fig. 3D). In each cluster, the mutant with the highest stability was identified: T872I/T801I (NST-M2), S742K/D843E/A846K (NST-M3), I761L/Y864H (NST-M4), and R688K/M771S (NST-M5). Within each cluster, the mutations may not be mutually positive because of epistatic effects of single-point mutations. However, positional and correlation analyses revealed that the stability of NST-M8 (R688K/S742K/I761L/N771S/T801I/D842E/A846K/Y864H/T872L), obtained by summing the clusters, reflected additivity (Fig. 3E). Experimental results indicate that the stability of NST-M8 was enhanced approximately 11.3-fold compared with that of the NST-WT.

**Fig. 3 | Schematic representation of development pathway of robust *N*-sulfo-transferase. A** In step 1, we predicted 39 hot-spot amino acid sites through molecular dynamics simulation of glycosylation and multiple sequence alignment. Step 2 involved further analysis through PROSS engineering calculations and activity testing, resulting in the stable first-generation mutant NST-M1. In step 3, virtual saturation mutagenesis was applied to the remaining 24 hot-spot sites to exclude any unreasonable sites and mutations, identifying 15 sites for further optimization. Step 4 involved designing simplified degenerate codons for each of the remaining 15 mutation sites, and screening through mutation libraries. The beneficial single mutations obtained were iterated to achieve a robust eighth-generation mutant, NST-M8. **B** The specific enzyme activity of protein variants achieved through PROSS engineering. NST-design 3, featuring 15 single-point mutations compared with the wild-type enzyme, was validated and used for subsequent mutation rounds. The experiment was conducted with three independent experimental samples ($n = 3$). Data are presented as mean values ± SD. **C** The free energy changes obtained from virtual saturation screening. Residues marked in blue on the left-axis indicate hit amino acid sites, while red signifies unsuitable mutation sites. Dashed-red boxes represent the amino acid hits, and refined degenerate codon libraries were constructed to generate the corresponding protein mutants for activity screening. **D** The stability of protein variants determined

through virtual saturation screening. Blue on the left-axis represents hit amino acid sites, while red indicates unconsidered mutations. Enzyme activity was determined by incubating enzyme at 50 °C for 0.5 h, followed by residual enzyme activity measurement at 37 °C. The activity of transferring 1 μmol of sulfate from the donor to the receptor per hour was defined as 1 IU, with units μmol/h. The experiment was conducted with three independent experimental samples ($n = 3$). Data are presented as mean values ± SD. **E** The stability enhancement process of mutants in iterative evolution involves obtaining the most stable mutants through the combination of a combinatorial library selection. The optimal mutations from each group were as follows: Group 1: NST-M2 (T872L/T801I); Group 2: NST-M3 (S742K/D843E/A846K); Group 3: NST-M4 (I761L/Y864H); and Group 4: NST-M5 (R688K/N771S). These mutations were further combined to obtain mutants with enhanced stability. NST-M2 and NST-M3 were combined to form NST-M6 (T872L/T801I/S742K/D843E/A846K), while NST-M4 and NST-M5 were combined to form NST-M7 (I761L/Y864H/R688K/N771S). Finally, NST-M6 and NST-M7 were combined to obtain the most stable mutant, NST-M8 (T872L/T801I/S742K/D843E/A846K/I761L/Y864H/R688K/N771S). The experiment was conducted with three independent experimental samples ($n = 3$). Data are presented as mean values ± SD. Source data are provided as a Source Data file.

## PROSS-FRISM design leads to robust *N*-sulfotransferase

NST-M8 was generated by reconstruction of the WT enzyme, resulting in a stable rigid protein scaffold. Employing NST-WT (PDB: 1NST) as the template, a structural model of the mutant M8 was derived by homology modeling. The designed mutations in NST-M8 were dispersed throughout the enzyme, showcasing typical features of stable mutations, such as increased loop rigidity (S637P, S741P, E839P, T869P), enhanced surface polarity (G625S, S637P, E660D, R688K, V699I, D721N, S738D, S742K, K743E, I761L, A767Y, N771S), promoted helix capping (E839P, L842E), and optimized core packing (D843E, A846K, Y864K, T869P, T872L) (Fig. 4A). NST-M8 exhibited greater surface polarity and increased rigidity compared with the WT, leading to enhanced overall stability of the protein structure. This suggests that the protein engineering applied to NST in *E. coli* followed an evolutionary trajectory of protein performance repair. This approach avoided disrupting the wild-type protein structure by avoiding deeply penetrating the core, thereby eliminating the need to consider the stability–activity trade-off.

Gradual evolution along the evolutionary trajectory leads to incremental enhancements in thermal stability and activity[52]. The most thermally stable and active NST variant, NST-M8, was mutated in 24 amino acids compared with the NST-WT. Compared with the NST-WT, NST-M8 also exhibited a 2.6-fold increase in enzyme activity (Fig. 4B), accompanied by a rise in the melting temperature from 54 °C to 68 °C (Supplementary Fig. 11). Stability experiments revealed that the WT enzyme retained its activity at 37 °C for no more than 1 day, while NST-M8 maintained approximately 50% of its activity even after 7 days. Immobilization on ReliSorb SP400 carrier via a Zbasic2-tag was employed to protect NST-M8 from environmental factors. The immobilized Z-NST-M8 maintained >90% activity after 9 days at 37 °C (Fig. 4C). This immobilization not only enhanced the enzyme stability but also enabled multiple reuses, lowering costs and improving efficiency. These advantages position immobilized NST-M8 for a broad range of applications in industrial production and biotechnological processes.

We also evaluated the performance improvement of NST-M8 through the *N*-sulfation reaction of the pentasaccharide-desulfated heparin synthesized using the chemoenzymatic method. Over 95% of the substrate reacted at a concentration of 24.58 g/L, which is four times the previous concentration limit for the NST-WT (6.14 g/L). The enzyme activity increased from 505 IU/L to 1280 IU/L. The catalytic efficiency ($k_{cat}/K_m$) increased from 3.76 μM/s to 5.55 μM/s. In terms of synthesis capacity, NST-M8 produced 4.84 g of *N*-sulfated pentasaccharide per liter of LB medium, compared with only 0.69 g/L-LB for

NST-WT. In terms of stability, NST-WT had a half-life of only 0.42 days at 37 °C, while NST-M8 had a half-life exceeding 7.2 days (Fig. 4D). Ultimately, the preparation of kilogram-level *N*-site sulfated pentasaccharide required only 12 mg of NST-M8 protein (Supplementary Fig. 12). These results demonstrate the synthetic capacity of NST-M8.

The results demonstrate that the combination of computational design and experimental screening of <300 variants enabled the rapid engineering of NST variants with markedly enhanced thermal stability. The PROSS-FRISM approach proved to be effective, minimizing experimental workload while efficiently exploring promising routes between evolutionary cycles. By adopting knowledge-guided pathways, it effectively mitigates adverse effects of mutations, thereby overcoming apparent dead ends[53]. While a perfect endpoint of evolution may not be achieved through a limited exploration process alone[54], this strategy systematically increases the likelihood of introducing beneficial mutations at opportune times and regions with significantly decreased experimental effort. The computationally redesigned and evolved NST-M8, described above, exhibits substantial improvements in stability compared with the WT enzyme, providing an exciting platform for further protein engineering and evolution aimed at enhancing stability without compromising enzyme catalytic activity.

## Chemoenzymatic synthesis of bioengineered heparin polysaccharides

The development of high-stability NST mutants instilled confidence in our ability to synthesize bioengineered heparin with enhanced activity. *E. coli* strain K5ASSH was grown in a medium supplemented with extracellular GlcNAc. Well-grown cells were harvested, washed, and further cultivated in a medium containing Ac$_4$GlcNTFA and IPTG. The supernatant after centrifugation was mixed with three volumes of ice-cold ethanol for polysaccharide concentration. After treatment with a LiOH solution at pH 12 for 0.5 h, the solution pH was adjusted to 7.0. Subsequently, NST-M8 with PAPS was added for the *N*-sulfation reaction. The reaction product was loaded onto a DEAE anion exchange column for purification to obtain *N*-sulfated polysaccharides. The *N*-sulfated polysaccharides were then mixed with 0.5 mg/mL each of recombinantly expressed C5-epi[5], 2-OST[55], 6-OST[15], and 3-OST-1[56], as well as excess PAPS, to prepare K5ASSH-engineered heparin (K5EH). K5EH was also purified using DEAE anion exchange chromatography (Fig. 5A). High-performance gel permeation chromatography was employed to determine the molecular weight of K5EH, which was around 11 kDa before sulfation and 16.5 kDa after full sulfation (Supplementary Fig. 13). This indicates that mild alkaline hydrolysis did not

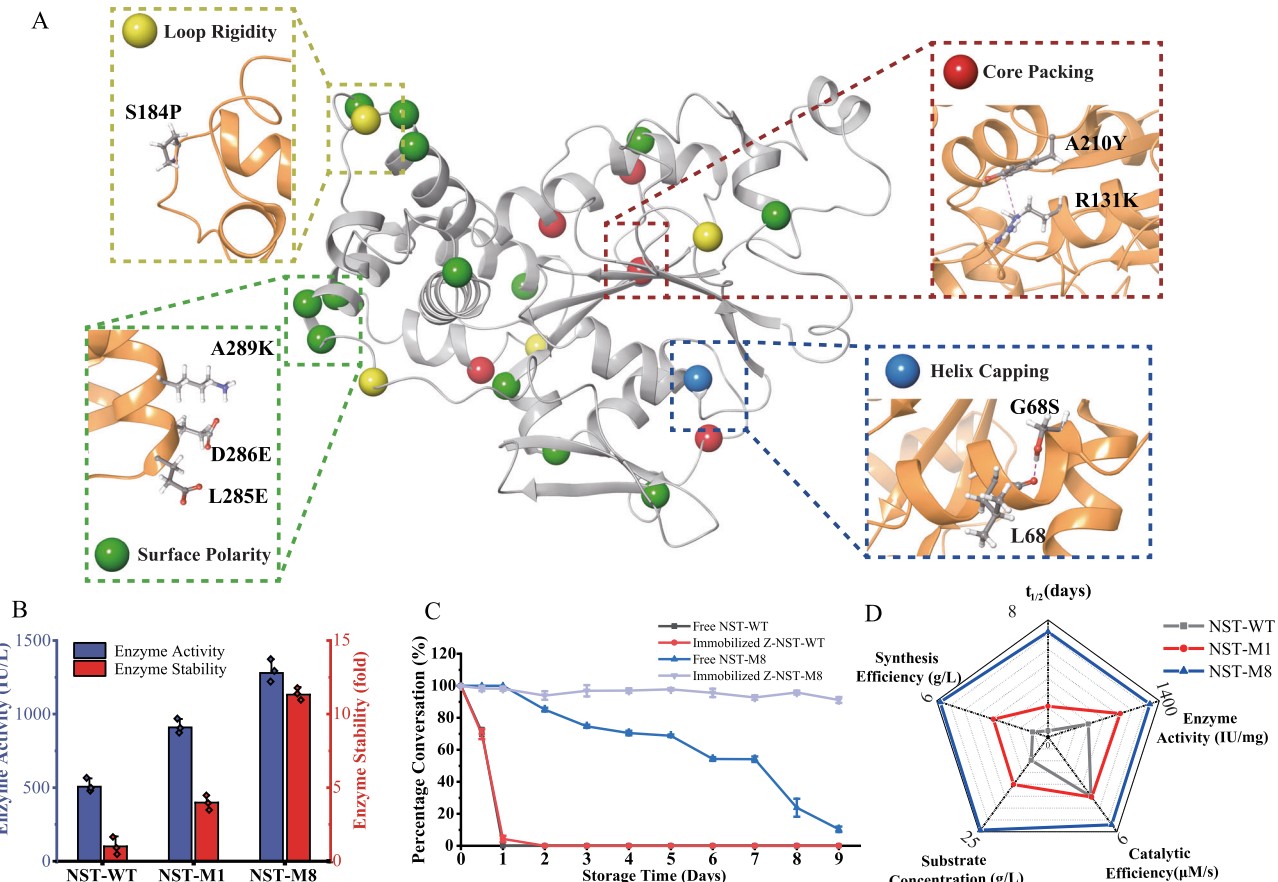

**Fig. 4 | Semi-rational design of a robust *N*-sulfotransferase mutant. A** The stable structural basis of the designed variant NST-M8 is depicted, with 24 mutated positions (compared with the wild-type enzyme) distributed throughout and represented as spheres. Spheres of different colors signify various reasons for stability enhancement. Yellow spheres indicate sites of stability improvement due to loop rigidity (S637P, S741P, E839P, T869P). Green spheres represent sites of stability improvement due to surface polarity (G625S, S637P, E660D, R688K, V699I, D721N, S738D, S742K, K743E, I761L, A767Y, N771S). Red spheres represent sites of stability improvement due to core packing (D843E, A846K, Y864K, T869P, T872L). Blue spheres represent sites of stability improvement due to helix capping (E839P, L842E). A thumbnail highlights the stabilizing effect of selected mutations. **B** The changes in enzyme activity and stability from wild-type NST to engineered mutant M1 and then to M8. Compared with the wild-type, M8 exhibited a 2.5-fold increase in activity and an 11.3-fold increase in stability. The experiment was conducted with three independent experimental samples ($n = 3$). Data are presented as mean values ± SD. **C** The stability of free and immobilized NST and its variants at 37 °C. NST-WT and NST-M8 was immobilized on ReliSorb SP400 carrier using enzyme surface tethering via a Zbasic2-tag. The experiment was conducted with three independent experimental samples ($n = 3$). Data are presented as mean values ± SD. **D** Radar plot showcases multiparameter progress half-life of enzyme at 37 °C (days), enzyme activity (IU/mg), catalytic efficiency(μM/s), substrate concentration(g/L), synthetic efficiency(g/L) from wild-type NST (gray) to NST-M1 (red) and NST-M8 (blue). Source data are provided as a Source Data file.

cause significant degradation of the polysaccharide. To elucidate the relationship between the composition and activity of K5EH, we conducted disaccharide analysis, and activity testing using unfractionated heparin (UFH), three low-molecular-weight heparins (LMWHs; enoxaparin, nadroparin, dalteparin), fondaparinux, and dekaparin (a chemoenzymatically synthesized homogeneous anticoagulant dodecasaccharide) as comparators (Fig. 5B). Meanwhile, all heparin anticoagulant factor (Xa and IIa) inhibitory activities were measured using fluorescent substrates (Fig. 5C).

K5EH exhibited a high degree of sulfation, with disaccharide analysis revealing that, except for the heparan sulfate (HS) component which had a lower degree of sulfation, all other components of ΔUA2S-GlcNS6S reached >60% sulfation (Supplementary Table 4). The average composition of 2.64 sulfate groups per disaccharide in K5EH was higher than that in UFH and LMWH (2.16–2.40 sulfate groups/disaccharide), but lower than in the homogeneous heparin drugs fondaparinux sodium (3.20 sulfate groups/disaccharide) and dekaparin (2.83 sulfate groups/disaccharide) (Table 1). Considering the importance of deacetylation of the *N*-position in heparin synthesis, the high level of sulfation in K5EH indicates the efficiency of

GlcNTFA for preparation of the deacetylated heparin backbone in *E. coli* strain K5ASSH.

K5EH exhibits pharmacological activity comparable to that of LMWH. In anti-FXa and anti-FIIa activity assays (Supplementary Fig. 14), the anti-FXa activity of K5EH reached 246.09 IU/mg, 1.3-fold that of UFH, while the anti-FIIa activity was only 0.26 times that of UFH (Table 2). In contrast, the homogeneous heparin drugs fondaparinux and dekaparin exhibit high anti-FXa activity but lack anti-FIIa activity. The ratio of anti-FXa to anti-FIIa activity of K5EH reached 5.06, akin to the ratio observed for LMWH (approximately 3–5), rather than UFH (approximately 1.0). The results of the half-maximal inhibitory concentration ($IC_{50}$) for coagulation factors showed that K5EH exhibited an $IC_{50}$ of 30.58 ± 3.84 ng/mL of anti-FXa Activity (Fig. 5D). K5EH demonstrated similar anti-FXa properties to fondaparinux and dekaparin. However, in terms of anti-FIIa activity, the $IC_{50}$ curve for K5EH was 157.99 ± 14.67 ng/mL, indicating similarity to that of low-molecular-weight heparin (Fig. 5E). We hypothesize that the presence of more sulfate groups, due to more extensive deacetylation of GlcNTFA, leads to higher anticoagulant activity. Additionally, although disaccharide analysis cannot fully identify disaccharide units

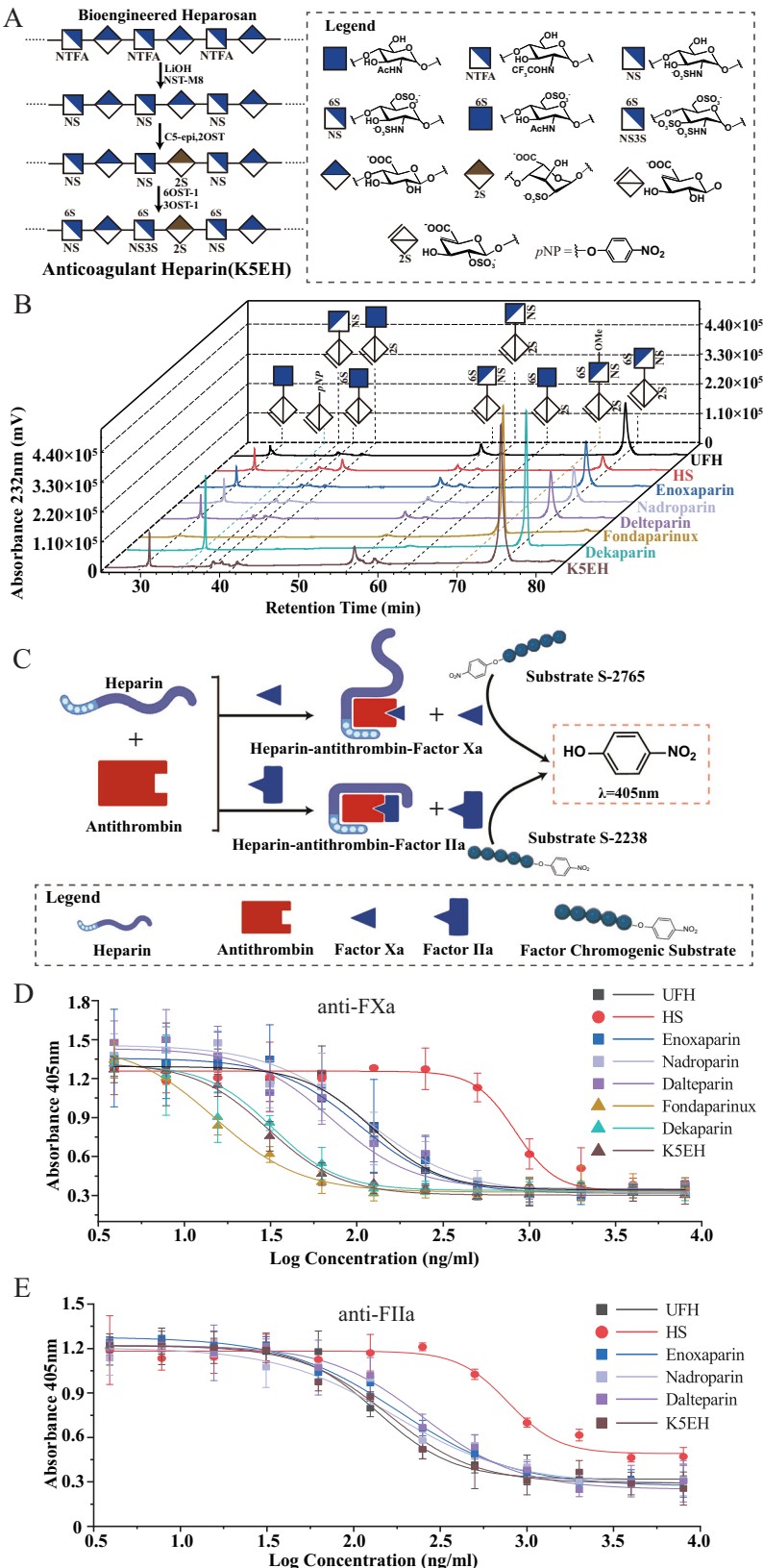

containing glucosamine 3-*O*-sulfation[57,58], we have reason to speculate that the high efficiency of the 3OST-1 recombinant enzyme we used results in higher anti-FXa activity, because 3-*O*-sulfation is essential for anti-FXa anticoagulant activity. This indicates that a higher level of sulfation will lead to increased anti-FXa activity. The elevated anti-FXa/FIIa activity ratio of K5EH suggests a decreased risk of bleeding during use, akin to LMWH. These findings underscore the viability of enzymatic synthesis and modification of heparin as an alternative to animal-derived heparin.

The presence of GlcNS residues is crucial for all subsequent modifications in the biosynthesis of HS. An ex vivo collaboration between *E. coli* strain K5ASSH and NST-M8, coupled with a

**Fig. 5 | Structural analysis and anticoagulant activity assays of bioengineered heparin polysaccharides. A** The synthesis process of K5 engineered heparin. The *N*-site sulfated polysaccharides prepared are used to synthesize anticoagulant K5 engineered heparin through a multienzyme cascade system involving C5-epi, 2OST, 6OST−1, and 3OST−1. In this process, C5-epi catalyzes the isomerization reaction of GlcA at the C5 position, leading to the formation of IdoA residues. Subsequently, 2OST catalyzes the sulfation reaction at the 2- position of IdoA, while 6OST-1 facilitates the sulfation reaction at the 6- position of GlcNS. 3OST-1 mediates the sulfation reaction at the 3- position of GlcNS6S. **B** PAMN-HPLC analysis of disaccharides resulting from HepI, HepII, and HepIII degradation of K5EH. The retention times for various disaccharides were: ΔUA-GlcNAc, -30.5 min; ΔUA-*p*NP, -34.5 min; ΔUA-GlcNS, -36 min; ΔUA-GlcNAc6S, -38 min; ΔUA2S-GlcNAc, -41 min; ΔUA-GlcNS6S, -56 min; ΔUA2S-GlcNS, -58.5 min; ΔUA2S-GlcNAc6S, -63 min; ΔUA2S-GlcNS6S-OMe, -69.5 min; and ΔUA2S-GlcNS6S, -74 min. The heparins used included: unfractionated heparin (UFH), heparin sulfate (HS), three low-molecular-weight heparins (Enoxaparin, Nadroparin, and Dalteparin), Fondaparinux (chemically synthesized heparin pentasaccharide), Dekaparin (chemoenzymatically synthesized low-molecular-weight heparin), and K5EH (heparin prepared in this study

using *E. coli* strain K5ASSH, NST-M8, and other sulfotransferases). **C** Detection principle of anticoagulant factor activity. After binding with antithrombin, heparin enhances its association with coagulation factors Xa or IIa. The binding to FXa requires a specific pentasaccharide sequence, while the binding to FIIa requires a glycan chain structure of >18 sugar residues in length. Subsequently, a specific fluorescent substrate is added to detect the remaining coagulation factors (S-2765 for FXa, S2238 for FIIa). The coagulation factors hydrolyze the fluorescent substrate, releasing *p*-nitrophenol, which is then detected at 405 nm. **D** Anti-FXa activity assays. FXa activity was determined by the rate of absorbance increase of *p*-nitrophenol at 405 nm. Each data point represents the average of three determinations. Data are presented as mean values ± SD. **E** Anti-FIIa activity assays. FIIa activity was determined by the rate of absorbance increase of *p*-nitrophenol at 405 nm. Each data point represents the average of three determinations. Data are presented as mean values ± SD. C5-epi *Homo sapiens* heparin C5 epimerase, 2OST *Gallus gallus* heparin 2-*O*-sulfotransferase, 6OST-1 *Mus musculus* heparin 6-*O*-sulfotransferase isoform 1, 3OST-1 *Homo sapiens* heparin 3-*O*-sulfotransferase isoform 1, IdoA Iduronic acid. Source data are provided as a Source Data file.

## Table 1 | Catalytic properties for wild-type NST (NST-WT) and NST-M8

| | $K_m$ (µM) | $K_{cat}/K_m$ (µM$^{-1}$·s$^{-1}$) | Enzyme activity (IU/mg) | Purification yield (mg/L) | $t_{1/2}$ (day) | $T_m$ (°C) | Substrate concentration (g/L) | Synthetic efficiency (g/L) |
|---|---|---|---|---|---|---|---|---|
| NST-WT | 100.47 ± 2.74 | 378.43 ± 0.46 | 504.67 ± 12.62 | 11.20 ± 0.37 | 0.42 | 54 | 6.14 | 0.69 ± 0.10 |
| NST-M8 | 133.43 ± 2.02 | 739.57 ± 3.26 | 1280.33 ± 23.76 | 50.80 ± 3.38 | 7.2 | 66 | 24.58 | 4.84 ± 0.54 |

Kinetic measurements at limiting concentrations of the acceptors pentasaccharide GlcA-GlcNH$_2$-GlcA GlcNH$_2$-GlcA-*p*NP (0.5 mM) but a variable concentration of the donor substrate PAPS (0.05 to 1 mM). $t_{1/2}$ (days) is the half-life of the enzyme at 37 °C. $T_m$ (°C) is the protein melting temperature determined by differential scanning fluorimetry. Substrate concentration (g/L) refers to the maximum tolerated concentration of acceptor pentasaccharide substrate by the enzyme. Synthetic efficiency (g/L) is the amount of *N*-sulfated pentasaccharide that can be synthesized using 1 L of LB medium expressing the NST enzyme. Three independent samples were used for $K_m$, enzyme activity, purification yield, and synthetic efficiency calculations. Data are presented as mean value ± SD. Source data are provided as a Source Data file.

## Table 2 | Comparison of characteristics of different heparins

| Heparin | Anti-FXa | | Anti-FIIa | | Anti-FXa/ Anti-FIIa | Molecular weight | Average sulfation per disaccharide |
|---|---|---|---|---|---|---|---|
| | IC$_{50}$ (ng/mL) | Potency (IU/mg) | IC$_{50}$ (ng/mL) | Potency (IU/mg) | | | |
| UFH | 123.16 ± 7.35 | 185* | 137.99 ± 8.44 | 185* | 1.00 | ~14000 | 2.40 |
| Heparan sulfate | 827.91 ± 137.77 | <2.00 | 779.84 ± 73.13 | 2.73 ± 0.60 | / | ~6000 | 1.62 |
| Enoxaparin | 104.04 ± 11.81 | 110.55 ± 21.26 | 182.41 ± 16.02 | 28.09 ± 7.51 | 3.94 | ~4500 | 2.29 |
| Nadroparin | 109.30 ± 22.37 | 131.86 ± 8.17 | 171.77 ± 26.22 | 30.31 ± 5.42 | 4.30 | ~4500 | 2.16 |
| Dalteparin | 71.10 ± 18.20 | 123.97 ± 19.23 | 251.84 ± 29.37 | 31.32 ± 6.55 | 3.96 | ~5500 | 2.36 |
| Fondaparinux | 14.99 ± 2.80 | 631.42 ± 12.35 | N.D | N.D | / | 1754 | 3.20 |
| Dekaparin | 32.39 ± 2.42 | 499.46 ± 20.57 | N.D | N.D | / | 3521 | 2.83 |
| K5EH | 30.58 ± 3.84 | 246.09 ± 8.69 | 157.99 ± 14.67 | 48.62 ± 3.96 | 5.06 | ~16500 | 2.64 |

Note: IC$_{50}$, potency, molecular weight, composition, and anti-FXa and anti-FIIa activities of heparin drugs. The experiment was conducted using three independent samples to measure IC$_{50}$ and potency. Data are presented as mean value ± SD. The molecular weight of K5EH was determined using high-performance gel permeation chromatography, while the molecular weights of other heparins were obtained from the supplier's instructions. *Unfractionated heparin (UFH) with anti-FXa and anti-FIIa activity of 185 IU/mg for each was used as the standard. N. D means not detected. Source data are provided as a Source Data file.

chemoenzymatic approach, effectively accomplished *N*-site sulfation of bioengineered heparosan, culminating in the synthesis of anticoagulant heparin. This semisynthetic strategy, rooted in a cellular system, has significant promise for the scalable production of bioengineered heparin.

## Discussion

Heparin is an important anticoagulant drug widely used in medical procedures to prevent the formation of blood clots[2]. This study demonstrates the synthesis of anticoagulant heparin using a semisynthetic strategy. This process involves in vitro cooperation between the *E. coli* strain K5ASSH and NST mutant NST-M8 for the *N*-site sulfation of bioengineered heparin polysaccharides. The presence of GlcNS residues is crucial for all subsequent modifications in the biosynthesis of heparin sulfate (HS), and this approach effectively

manipulates these residues to enable the generation of bioengineered heparin.

In the disaccharide repeat units of the capsule polysaccharides synthesized using our method, a non-natural analog of *N*-acetylhexosamine efficiently and site-specifically substitutes in heparin, allowing the reliable and predictable introduction of functional groups onto bacterial surfaces. Chemical *N*-deacetylation and *N*-sulfation reactions are typically included in chemoenzymatic heparin synthesis, wherein heparosan is deacetylated using an aqueous solution of NaOH. Subsequently, the *N*-sulfation step is conducted using trimethylamine-trioxide sulfur to prepare the intermediate *N*-sulfated heparosan polysaccharide[59]. Excessively strong alkaline conditions can cause chain depolymerization reactions, hence strict control of the chemical reaction conditions is necessary to regulate the resulting *N*-sulfated heparin derivatives[60]. The analog GlcNTFA can be efficiently

incorporated into capsule polysaccharides. The engineered K5 polysaccharide so obtained was subjected to mild enzymatic synthesis for *N*-sulfation of the polysaccharide, avoiding the need for harsh chemical reaction conditions and the use of difficult-to-recombinantly-express NDST proteins[61].

Through PROSS-FRISM engineering, we gradually developed the NST-M8 protein along an evolutionary trajectory through a series of mutations. Although this is a limited exploration process that may not achieve a perfect evolutionary endpoint, this strategy systematically increases the possibility of introducing beneficial mutations at the right time and region, greatly reducing the experimental workload[62]. As described above, the computationally redesigned and evolved NST-M8 exhibits significant improvement in stability compared to the WT enzyme, providing an exciting platform for further protein engineering and evolution. Within the heparin synthesis system, NST represents the simplest sulfotransferase, comprising only 325 amino acids. However, due to the sulfotransferase activity characteristics of the enzyme and the lack of cell-based high-throughput screening methods for sulfotransferase, there is a need for knowledge-guided strategies to reduce the screening throughput of sulfotransferase. Therefore, NST serves as an excellent example for research and design within the PROSS-FRISM strategy. Future research can focus on the semi-rational design of other heparin-modifying enzymes, such as C5-epi, 2-OST, 6-OST, and 3-OST. By employing the PROSS-FRISM strategy, the stability and catalytic efficiency of these enzymes can be enhanced, leading to the development of robust industrial enzyme preparations. The application of the PROSS-FRISM strategy to expand the utility of these recombinant heparin-modifying enzymes will offer greater opportunities in the field of heparin-related research and applications.

Furthermore, in protein development, particularly enzymes, the trade-off between stability and activity is an important aspect to consider. This trade-off refers to the balance between the stability of the enzyme, or its ability to maintain its structure and function under different conditions, and the activity, or its ability to catalyze reactions. The increased thermal stability of NST-M8 allows it to maintain its activity at higher temperatures and for longer periods of time compared to the WT enzyme. This is crucial as enzymes often lose functionality at high temperatures. Additionally, the increased activity of NST-M8 enables more efficient catalysis of reactions, as demonstrated by its enhanced performance in the *N*-sulfation reaction of heparin. Immobilized enzymes further improve their stability, protecting them from environmental factors and allowing for repeated use. This not only prolongs the lifespan of the enzyme but also reduces costs, improves efficiency, and makes it a valuable tool for industrial production and biotechnological processes.

Finally, enzymatic chemical modification of engineered K5 polysaccharides resulted in K5EH. The results demonstrate that K5EH has a higher anti-FXa to anti-FIIa ratio, indicating a higher degree of sulfation. Sulfation is a crucial process for the biological activity of heparin[14], and this higher level of sulfation increases anti-FXa activity. Anti-FXa targets the key enzyme factor Xa in the coagulation cascade, inhibiting its function and preventing blood clot formation[63]. On the other hand, anti-FIIa targets another key participant in coagulation, thrombin. Increasing the ratio of anti-FXa to anti-FIIa is beneficial as it provides effective anticoagulation while minimizing the risk of bleeding, which is a common adverse reaction to heparin therapy.

The chemoenzymatic approach is an effective method for preparing structurally defined GAG oligosaccharides that are difficult to synthesize using traditional chemical synthesis methods. This method is suitable for specific biological and pharmacological applications[17,26], representing the future direction of the heparin industry. Non-animal sourced heparin is the primary challenge that the heparin industry must address, to replace current production lines of UFH and low-molecular-weight heparin (LMWH). The characteristics of K5EH, along with its pharmacological activity comparable to another widely used anticoagulant, LMWH, suggest its potential efficacy as a heparin substitute. The use of bioengineered heparin with a higher anti-FXa/FIIa activity ratio can significantly improve the safety and efficacy of heparin therapy, indicating a reduced risk of bleeding during its administration. We emphasize the feasibility of enzymatic synthesis and modification of heparin as an effective and scalable alternative to animal-sourced heparin. It opens up new avenues for producing safer and more reliable anticoagulant drugs, addressing ethical and supply chain issues associated with traditional heparin. Future research should focus on optimizing this semi-synthetic strategy and testing the safety and efficacy of K5EH in a clinical setting.

## Methods

### Synthesis of Ac$_4$GlcNTFA

1,3,4,6-tetra-*O*-acetyl-*β*-D-glucosamine hydrochloride (2.04 g; Samuel, Jinan, China) was mixed with anhydrous sodium carbonate (1.00 g) and dissolved in 10 mL of anhydrous methanol; 3.5 mL of ethyl trifluoroacetate (Sigma) was added. The mixture was stirred at room temperature for 12 h. The residue was purified by flash column chromatography (gradient cyclohexane/ethyl acetate v/v 3:1–1:1) to obtain 2.32 g of product with a yield of 92%.

### Preparation of *E. coli* strain K5 polysaccharide analog

The K5ASSH strain[32] of *E. coli* was cultured with chloramphenicol (34 μg/mL), kanamycin (50 μg/mL), and GlcNAc (100 μg/mL). Cells were activated in liquid LB medium at 225 rpm and 37 °C for 12 h. After consuming GlcNAc for 1 h, Ac$_4$GlcNTFA (500 μg/mL) was added to the culture medium and cultivation was continued at 37 °C and 225 rpm until OD$_{600}$ (the optical density at 600 nm) reached 0.6. At that time, IPTG was added to the culture medium (to 0.2 mM). Cultivation was continued at 22 °C for 12 h. After centrifugation to remove bacteria, the 3-times volume of ethanol was added to the supernatant, which was precipitated overnight at 4 °C. The crude polysaccharide obtained was purified by using a Bio-Gel P-10 column to yield the K5 polysaccharide analog.

### Analysis of K5 polysaccharide analog

After treating 1 mg/mL of K5 polysaccharide analog in 0.1 M LiOH for 1 h, 50 mM MES (4-morpholine ethanesulfonic acid) was added, and the pH was adjusted to approximately 7.0. Subsequently, 0.2 mg/mL NST and 2 mg/mL PAPS were added, and reaction was carried out at 37 °C for 12 h before purification of polysaccharide using a DEAE column. This reaction converted all -GlcNTFA-GlcA- into -GlcNS-GlcA-. Subsequently, 0.1 mg/mL Hep III[64] and 5 mM CaCl$_2$ were added to the reaction overnight, and the product was analyzed by PAMN-HPLC. The lysed disaccharide product was purified by DEAE column chromatography and desalinated by using a Bio-Gel P-2 column before being subjected to HPLC-MS analysis.

### MD simulations

The crystal structure of *N*-sulfotransferase (PDB: 1NST) was meticulously refined using Schrödinger. Additionally, the GlcNAc$_2$Man$_3$ pentasaccharide structure, extracted from the human C5-epimerase crystal structure (PDB: 6HZZ[49]), was carefully grafted onto the *N*-linked site (residue N667) of NST. This procedure enabled the simulation of the glycosylated form of the wild-type *N*-sulfotransferase (NST) structure. We conducted Molecular Dynamics (MD) simulations using the advanced Amber software suite. The Amber ff14SB force field was specifically employed for the *N*-sulfonyltransferase, in both its glycosylated and non-glycosylated states[65]. The respective force field parameters for the ligand were generated using the Amber antechamber module in collaboration with the General Amber Force Field (GAFF)[66]. We immersed the protein structure in a truncated octahedral box, which extends 8 Å beyond the solute, and used the TIP3P water model while applying periodic boundary conditions. The MD simulations,

lasting for 20 ns, were performed under the NPT (Number of Particles, Pressure, Temperature) condition. All MD results were subsequently analyzed using the ptraj module, an indispensable part of the Amber software suite. The variation in root-mean-square fluctuation (RMSF) value due to the presence or absence of glycosylation was calculated as ΔRMSF.

## Multiple sequence alignment

A search of the amino acid sequence HMM (Hidden Markov Model) was conducted using the HMMER webpage (http://hmmer.org/). The alignment was subsequently employed to construct a HMM profile, courtesy of the "hmmbuild" function. For the purpose of visualization, the HMM logo was forged via the Skylign online tool[67] (http://skylign.org/). The frequencies of each letter in the designated target position were calculated, founded on the relative entropy or Kullback-Leibler distance.

## PROSS engineering of wild-type NST

In identifying candidate amino acid residues for mutagenesis, to ensure reliability while covering a comprehensive array of residues, a Δroot-mean-square fluctuation value of <0.6 in the MD simulations before and after glycosylation, along with a score of <8 for the residue in sequence conservation analysis (values range from 0 to 10, with 10 indicating complete amino acid conservation at that position) were applied as thresholds. A total of 39 amino acids were hit: G625, S637, E660, T669, D682, V685, R688, A691, K698, V699, T701, N705, D721, S738, S741, S742, K743, I761, A767, Y768, H769, A770, N771, Q772, K779, E784, K787, M791, T801, H805, T807, G823, E839, L842, D843, A846, Y864, T869, and T872.We used the Protein Repair One Stop Shop (PROSS)[42]. server, a bioinformatics strategy, to design a series of mutants with increasing surface polarity. The crystal structure of NST (1NST) served as the template for the PROSS algorithm. It is worth noting that only above 39 specific sites were targeted for engineering, which were identified through dynamic simulations and multiple sequence alignment. The algorithm suggested potential amino acid substitutions based on sequence homologs that shared at least 75% similarity with the template 1NST. None of these substitutions was located within the active or substrate-binding sites of the enzyme. From the suggested mutants, we selected four for expression named NST-design 1-4.

## Virtual saturation mutation calculation

After activity testing, the most stable NST-design 3 (NST-M1) include 15 mutations: G625S, S637P, E660D, V699I, D721N, S738D, S741P, K743E, A767Y, K779Q, E784D, T807H, E839P, L842E, and T869P. Out of the original 39 candidate sites, there are 24 remaining sites: T669, D682, V685, R688, A691, K698, T701, N705, S742, I761, Y768, H769, A770, N771, Q772, K787, M791, T801, H805, G823, D843, A846, Y864, and T872. Based on the crystal structure of NST (PDB ID: 1NST), energy calculations with Rosetta_ddg[47] were performed; these 24 sites were mutated in silico to all proteinogenic amino acids.

## Preparation of point mutant library using one-step PCR method

Construction of the point mutant library was carried out using the Vazyme Mut Express II Fast Mutagenesis Kit V2, and specific operations can be found in the instruction manual. In short, PCR amplification of the NST-M1-encoding plasmid was performed using specific primers with simplified codons. The selection of simplified codons is shown in Fig. 3C. The amplified product was digested by DpnI and cyclized by ClonExpress recombination, and then directly transformed into DH5a to complete site-specific mutagenesis.

## Preparation of combinatorial mutation library through gene recombination

The Mut Express MultiS Fast Mutagenesis Kit V2 (Vazyme, Nanjing, China) was used for the construction, and specific operations can be found in the instruction manual. In brief, the template plasmid (pGEX-4T-1-NST-M1) was divided into three segments based on the target mutation sites. Partially complementary primer mixtures (including mutant and wild-type genotypes) were designed for each target mutation site. Using the template plasmid (pGEX-4T-1-NST-M1) as a template, amplification and recovery were performed separately for each segment. After digestion with DpnI, the amplified products were subjected to recombination and ligation. The recombined products were directly transformed into DH5a to complete the construction of the mutant library. All primers used for this step can be found in Supplementary Data 1.

## General methods for purification of NST proteins

The NST (UniProt P52848.1, residues L557–R882) and its variants were genetically engineered into vector pGEX-4T-1[61] (GE Healthcare), before being chemically introduced into E. coli Origami B (DE3) cells (Novagen). The transformed cells were incubated with shaking at 225 rpm at 37 °C for 4 h, until $OD_{600}$ reached approximately 0.8; then, IPTG was added to a final concentration of 0.2 mM. Protein expression was induced at 22 °C and 225 rpm for 16–18 h. The NST was purified using a glutathione S-transferase column supplied by Sangon Biotech (Shanghai, China).

## General method for determining NST stability by PAMN-HPLC

The experiment involved the use of deacetylated heparin pentasaccharide GlcA-GlcNH$_2$-GlcA-GlcNH$_2$-GlcA-pNP[12] as the acceptor substrate and PAPS as the donor substrate. Initially, the enzyme was subjected to heat treatment at 50 °C for 0.5 h. The reaction system was comprised of 50 mM MES (pH 7.0), 0.2 mM pentasaccharide acceptor substrate, 0.6 mM PAPS, and 0.05 mg/mL protein. The reaction was conducted in a water bath, maintained at 37 °C, for 0.5 h, followed by inactivation of the enzyme by boiling for 5 min. The reaction solution was filtered using a 0.22-μm membrane and subjected to PAMN-HPLC detection. The pNP group of the monosaccharide acceptor showed specific absorption at 310 nm. One IU of enzymatic activity was defined as the amount that transferred 1 μmol of sulfate to the acceptor substrate per hour (unit, μmol/h).

## General method for determining NST stability using microporous plates

Synthesis of primers for mutation was performed by Genscript (Nanjing, China), and the mutation library was constructed using the above methods. Recombinant plasmids were chemically transformed into E. coli Origami B (DE3) cells. A single colony was picked and placed in 1 mL of sterilized LB medium containing 50 μg/mL carbenicillin. This was then cultured at 37 °C and 225 rpm. The overnight culture broth was transferred to 2 mL of fresh LB medium (2% v/v inoculum); the newly inoculated culture was incubated at 37 °C, 225 rpm, for approximately 4 h until $OD_{600}$ reached approximately 0.8. Then, 0.2 mM IPTG was added, and protein expression was induced at 22 °C and 225 rpm for 16–18 h. The bacterial cells were collected by centrifugation, washed twice with phosphate-buffered saline (PBS), and subjected to two cycles of freezing and thawing. The cellular material was then resuspended in PBS that contained 0.1 mg/mL lysozyme and incubated at 37 °C for 1 h. After centrifugation, the supernatant was collected. A standard reaction system was set up that included 50 mM MES (pH 7.0), 0.2 mM pentasaccharide receptor substrate, 0.6 mM PAPS, 2 mM pNPS, 0.2 mg/mL ASTIV, and 5 μL supernatant. The absorption at 410 nm was then measured at 37 °C for 90 min. The screening results of all mutants can be found in Supplementary Data 2.

## Enzyme immobilization

A total of 60 mg ReliSorb SP400 carrier (Biokal, Netherlands) was weighed out and put in an Eppendorf tube. The carrier was washed three times with water, followed by two washes with PBS.

Approximately 1 mL of purified NST-WT or NST-M8 protein was added and incubated at room temperature with agitation at 40 rpm for 2 h. The supernatant was removed and the carrier was washed with PBS. The enzyme fixation was analyzed by SDS-PAGE. NST or immobilized NST was placed at 37 °C for 1–9 days; enzyme activity determination was performed according to the general method for determination of NST activity by PAMN-HPLC.

### Differential scanning fluorimetry determination of protein melting temperature

The thermal denaturation assay using differential scanning fluorimetry (DSF) was conducted for NST-WT and NST-M8. Protein samples were diluted to a final concentration of 5 μM in a 20 mM buffer with varying pH values and 150 mM NaCl. DSF data was collected using a CFX96 RT-PCR instrument (Bio-Rad) with the "HEX" channel for fluorescence excitation and emission. The protein samples were dispensed into a 96-well Frame Star PCR plate and covered with a clear thermal-seal film to prevent evaporation. The temperature was increased in increments of 0.5 °C with a 5-second equilibration hold at each temperature. As the temperature increased, the protein unfolded, exposing its hydrophobic core to the solvent. The hydrophobic sites were bound by SYPRO Orange dye (Sigma), leading to an increase in fluorescence. Fluorescence intensity was monitored at 570 nm, allowing for simultaneous and independent readings of all 96 wells. The fluorescence intensity data was fitted to a Boltzmann sigmoidal curve using Prism software to determine the melting temperatures of NST-WT and NST-M8.

### Preparation of bioengineered heparin

In previous work, we constructed expression strains for *Homo sapiens* heparin C5 epimerase (C5-epi[5]), *Gallus gallus* heparin 2-*O*-sulfotransferase (2OST[55]), *Mus musculus* heparin 6-*O*-sulfotransferase isoform 1 (6OST-1[15]), and *Homo sapiens* heparin 3-*O*-sulfotransferase isoform 1 (3OST-1[56]). All proteins were recombinantly expressed in, and purified from *E. coli*. -GlcNDFA-GlcA- polysaccharide analog was prepared according to the protocol "Preparation of K5 polysaccharide analogue". To prepare -GlcNS-GlcA- polysaccharide analog, 1 mg/mL of the K5 polysaccharide analog was treated with 0.1 M LiOH for 1 h. Then, 50 mM MES was added and the pH was adjusted to approximately 7.0. Subsequently, 0.2 mg/mL NST and 2 mg/mL PAPS were added, and reaction was carried out at 37 °C for 12 h. This reaction converted all -GlcNTFA-GlcA- structures into -GlcNS-GlcA-. After the reaction, purification was performed using a DEAE column. For reaction involving 1 mg/mL of -GlcNS-GlcA- polysaccharide, 10 mg/mL PAPS and 0.2 mg/mL of each of C5-epi, 2OST, 6OST, and 3OST-1 were added, and reaction was carried out overnight at 37 °C. The product was then purified using a DEAE column and desalted using a Bio-Gel P-10 column. For analysis, a mixture of HepI, HepII, and HepIII was added to the engineered heparin, and then analysis of the disaccharide degradation products was performed using PAMN-HPLC.

### Determination of polysaccharide molecular weight

The molecular weight and purity of polysaccharides were determined using high-performance gel permeation chromatography. The chromatographic method employed a mobile phase of 0.2 M NaCl solution, a BRT105-103-101 series gel column (8 × 300 mm), a flow-rate of 0.8 mL/min, a column temperature of 40 °C, an injection volume of 25 μL, a refractive index detector RID-10A, and an analysis time of 60 min.

### Determination of anti-Xa and anti-IIa activity

The measurement of anti-Xa and anti-IIa activity was based on methods outlined in the Chinese Pharmacopoeia. Purified heparin was diluted stepwise to prepare solutions with a series of concentrations. Forty microliters of each concentration of the test sample solution was added to a 96-well enzyme-linked immunosorbent assay plate. Subsequently, 40 μL of antithrombin (ATIII; 1 IU/mL) was added to each well and incubated at 37 °C for 2 min. Then, 40 μL of Xa or IIa (8 μg/mL) was added and incubated at 37 °C for 2 min. Next, 40 μL of substrate S-2765 or S-2238 (0.8 mg/mL) was added and incubated at 37 °C for 2 min. Finally, 80 μL of 20% acetic acid was added to stop the reaction, and absorbance at 405 nm was measured immediately. A standard curve was established using an ungraded heparin standard. Nonlinear regression analysis was performed with absorbance as the y-axis and the concentration of the test sample solution as the x-axis to calculate $IC_{50}$ values for the inhibition of factor Xa or IIa by the heparin.

### Heparin disaccharide analysis

Unfractionated heparin, heparin sulfate, enoxaparin, nadroparin, dalteparin, and fondaparinux were purchased from Macklin. Dekaparin (chemoenzymatically synthesized low-molecular-weight heparin) was synthesized using chemoenzymatic method. Heparinases I, II, and III sourced from *Flavobacterium heparinum* were recombinantly expressed in *E. coli*. For disaccharide composition analysis of all heparins, a mixture of heparinases I, II, and III (each at 0.1 mg/mL) was used to completely digest samples overnight in 50 mM ammonium acetate buffer containing 2 mM $CaCl_2$. After boiling and centrifugation of the reaction mixtures (each initially containing 50 μg of heparin), the filtrate was subjected to HPLC analysis.

### HPLC for disaccharide composition analysis

PAMN-HPLC was used to analyze the disaccharide composition of heparin. The column was equilibrated with 1.8 mM sodium dihydrogen phosphate (mobile phase A, pH 3.0), followed by elution with a gradient of 1.8 mM sodium dihydrogen phosphate and 1 M potassium dihydrogen phosphate (mobile phase B, pH 3.0). The column temperature was maintained at 30 °C. The flow was set for a low-pressure gradient with a flow-rate of 0.5 mL/min. The gradient for mobile phase B was as follows: 0–6 min: 0%, 6–110 min: 0%–100%, 110–120 min: 0%; stopped at 120 min. The chromatograms were recorded using a UV detector at 232 nm. The injection volume was 10 μL. Disaccharide standards were purchased from Iduron (Manchester, UK).

### HPLC for NST activity analysis

The deacetylated heparin pentasaccharide GlcA-GlcNH$_2$-GlcA-GlcNH$_2$-GlcA-*p*NP was used as the substrate for NST. PAMN-HPLC was employed to analyze the reaction products. For PAMN-HPLC, the column used was a Polyamine II-HPLC, 4.6 × 250 mm, from YMC. The elution process involved a linear gradient of KH$_2$PO$_4$ from 0 to 1 M over 40 min, followed by 1 M KH$_2$PO$_4$ for 30 min, at a flow-rate of 0.5 mL/min. The chromatograms were recorded using a UV detector at 310 nm. The injection volume was 10 μL.

### Reporting summary

Further information on research design is available in the Nature Portfolio Reporting Summary linked to this article.

## Data availability

The sequences of all mutagenic primers used in this study are provided in Supplementary Data 1. The results of all screened NST mutant libraries are provided in Supplementary Data 2. Supplementary information. Source data are provided with this paper.

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

## Acknowledgements

This work was supported in part by the National Key R&D Program of China (2023YFA0914300), the Projects of Science and Technology Department of Shandong Province (Project no. 2022SFG0104 to JS) and Open Projects Fund of NMPA Key Laboratory for Quality Research and Evaluation of Carbohydrate-based Medicine and Shandong Key Laboratory of Carbohydrate Chemistry and Glycobiology.

## Author contributions

Conceptualization: J.-Z. S.; methodology: J.-Q. D., R.-M. X., Y.-J. W. and S.-Y. X.; investigation: J.-Q. D. and Y. L.; visualization: J.-Q. D., X.-Y. L. and Y.-L. C.; funding acquisition: J.-Z. S. and F.-S. W.; Supervision: J.-Z. S.; writing original draft: J.-Q. D.; writing review & editing: J.-Z. S. and J.-Q. D.

## Competing interests

The authors declare no competing interests.
