## [Peer Review File · Nature Communications]

Biosynthetic Production of Anticoagulant Heparin: Strategies for Engineering Efficient Sulfotransferases and Addressing Challenges with N-deacetylase/N-sulfotransferasesReviewers' Comments:

Reviewer #1:

Remarks to the Author:

This study successfully introduced GlcNTFA, an important precursor for heparin chemoenzymatic synthesis, into the capsular polysaccharides of *Escherichia coli* K5, namely heparosan. The key sulfation enzyme NST was also engineered with improved stability and activity. It could be important for cost-effective production of heparin or related polysaccharide compounds. The data looks robust, and the major conclusion is solid. However, how universally this approach could be applied to other enzymes is not addressed since only one enzyme was used as an example. Besides, other biosynthetic approaches of heparin based on heparosan or de novo have not been discussed thoroughly. I suggest a major revision before the manuscript could be published.

Major comments:

1. Fig 1C. What is the disaccharide pattern of heparosan without GlcNTFA feeding? Could there be any UA-GlcNH₂ signals? Suggest including a control chromatogram of natural heparosan disaccharides without GlcNTFA incorporation.
2. The enzyme evolution strategies have been largely explored and could be used either alone or in combination. This work employed the PROSS-FRISM platform to enhance the efficiency of NST. However, to generalize this approach, more work and discussion are required. Firstly, the explanation and schematic representation of this approach was not straightforward enough. The readers could not easily understand the importance and value of this approach. It seems the authors "rationally" picked several sites based on structural and modeling information and combined them with established PROSS and FRISM approaches. Maybe I did not completely understand the beauty of the semi-rational idea, the PROSS and FRISM approaches are beyond my expertise, but the authors could definitely do better to improve the manuscript to be more readable. Since only NST was studied as an example, it is probably challenging to extend to other enzymes. What is the situation of the other modifying enzymes (expression system, stability, activity, yield)? Can the other modifying enzymes like C5-epi, 2-OST, 6-OST, and 3-OST-1 be optimized similarly? Even though a workflow should at least be described for further optimization of other enzymes. More discussion about the current method for future applications is recommended.
3. The chemoenzymatic synthesis of heparin from heparosan has already been reported by several papers before. The author should cite those studies and compare and discuss the strategies. Is it possible to chemically incorporate GlcNS after the harvest of heparosan? Is the process in this study more cost-effective than previously published ones? Besides, the chemoenzymatic de novo synthesis of heparin oligosaccharides established by Jian Liu's group should be at least discussed in the manuscript. By comparing these approaches, in the author's opinion, what would be the future direction of production of heparin? More in-depth discussion was recommended.
4. How was the molecular size (length of sugars) of heparosan controlled? Is it controlled by metabolic engineering or chemical digestion? The MW distribution was not included in the analysis. (Table 2)

Minor comments:

Please carefully re-examine the manuscript for any typos and mistakes. Such as Fig.1A(E.coil for E.coli)

Reviewer #2:

Remarks to the Author:

Deng et al. present a chemo-enzymatic synthesis strategy for heparin, an important anticoagulant

drug. Animal-derived heparin is highly heterogeneous and it has led to substantial health risks in the past. Thus, there is a great ongoing effort in the glycobiology field to synthesize defined heparin molecules. Here, the authors describe a strategy in which they combine bacterial production of heparosan containing a synthetic GlcNAc analog with the in vitro sulfation of the polysaccharide backbone using recombinant enzymes. Two key advancements for carrying out this process are described: first, the use of the unnatural monosaccharide N-trifluoroacetylglucosamine (GlcNTFA) during heparosan production in *E. coli* K5 cells, which can subsequently be deacetylated chemically allowing to overcome the limitation of producing a deacetylase enzyme. A second advancement is the engineering of the human N-sulfotransferase (NSF) to increase its thermal stability and activity. The authors use the PROSS-FRISM platform to predict stability increasing mutations, which are then tested by extensive mutational studies of recombinant NST mutant enzymes. The experimental data on increased protein stability and anticoagulation activity looks convincing, however, the presented improvements might not reach the novelty bar needed for this journal.

Minor comments:

- description of figure 2B concludes that mutant NDST-M1 had highest expression level and stability, however, figure 2B shows a comparison of activities. How were the expression levels compared?

-What are the yields of purified NST-WT and NST-M8 protein?

- the results section is written lengthily and there are sections that would rather belong to the introduction (description of GlcNTFA strategy) or to the method section. It is hard to understand what are novel results and what is the description of previously established techniques

- L46-47 expression of NDST1 has recently also been described in mammalian expression systems (Vallet et al, 2023)

- L107-108 the authors state "we engineered *E. coli* K5 strain", but I don't see any author from this article in the corresponding reference "29". Other groups in the field should be acknowledged correctly.

Reviewer #3:

Remarks to the Author:

Manuscript by Deng et al introduces an enhanced cell-system based strategy for production of heparin (optimising N-Sulfotransferase activity using a rapid multi-method site directed mutagenesis strategy, PROSS-FRISM, using computational and experimental approaches). Mutagenesis screening resulted in dramatically increased activity of NST (11 fold), combined with improved stability (2.5 fold). When combined with semi-synthesis of substrate via mild weak base treatment of heparosan products containing GlcNAcAZ residues, a bioengineering process to make cell-based anticoagulant heparin was demonstrated, resulting in products with high anti-Xa anticoagulant activity (almost 250 IU/mg). High Xa/IIa activity was also noted, which could be beneficial in clinical use due to lower bleeding risk akin to that achieved by LMWHs. Some additional supportive technical advances are described, including a spectroscopic method for monitoring ST activity via conversion of PAPS sulfate donor pNPS (that allows more HT screening of NST and could be applied to other STS). The methodology appears sound and the data support the conclusions.

Collectively these studies represent a major advance and also demonstrate a strategy for improving enzymes catalytic function for the multi-enzyme cascade require to make highly complex heparin and

heparan sulfate products. This work could underpin more robust and efficient future production of novel therapeutics. Some minor revisions are need to render the work suitable for publication.

Minor comments

1. Fig 1A, misspelling E coli

2. Fig 2D, why are some mutants designated with a strikethrough ?

3. Fig 4D, incorrect symbol for K5EH, should be the brown triangle ??

4. Table 2: Is the molecular weight $\sim 30,000$ correct for UFH. Seems high since for most UFH samples it is around 12-15,000.

5. p11/12: the authors describe the high Xa/IIa anticoagulant activity of engineered K5EH (which is more akin to LMWH properties than UFH) and discuss possible reasons such as the higher sulfation compared to UFH. However, they should also mention that 3-O-sulfation, not measured in these products, could also be a factor, since this is required for anti-Xa but not IIa activity. Whilst not essential for the current publication, if the authors have data on 3-O-sulfation this should be included, or at least described as subject to future studies to characterise the products to underpin possible clinical development.

6. Some of the figure panels need to be prepared at higher resolution for better clarity.

Response to the reviewer reports on manuscript NCOMMS-23-63535

Dear Editors and Reviewers,

We are deeply appreciative of the constructive and insightful feedback provided by all three reviewers, as it has greatly assisted us in addressing key issues in this work. Enclosed is our detailed response (in **blue**) to all the comments raised by the reviewers. Revisions made to the main manuscript or the supplementary materials are highlighted in **red**.

■ Responses to the comments of Reviewer #1

This study successfully introduced GlcNTFA, an important precursor for heparin chemoenzymatic synthesis, into the capsular polysaccharides of Escherichia coli K5, namely heparosan. The key sulfation enzyme NST was also engineered with improved stability and activity. It could be important for cost-effective production of heparin or related polysaccharide compounds. The data looks robust, and the major conclusion is solid. However, how universally this approach could be applied to other enzymes is not addressed since only one enzyme was used as an example. Besides, other biosynthetic approaches of heparin based on heparosan or de novo have not been discussed thoroughly. I suggest a major revision before the manuscript could be published.

Response: Thank you for your valuable feedback and suggestions on our research. We appreciate your insights and will address the points raised in the revision.

Major comments:

1. Fig 1C. What is the disaccharide pattern of heparosan without GlcNTFA feeding? Could there be any UA-GlcNH₂ signals? Suggest including a control chromatogram of natural heparosan disaccharides without GlcNTFA incorporation.

Response: Thank you for your valuable suggestion. We have included an appropriate control chromatogram of natural heparosan disaccharides without GlcNTFA incorporation in **Figure 1C**. According to our liquid chromatography analysis results, there was no obvious UA-GlcNH₂ signal after disaccharide degradation of K5 polysaccharides from cells fed with natural GlcNAc (red). Similarly, after mild treatment with LiOH, there was no significant peak shape change in the K5 polysaccharides from cells fed GlcNAc. K5 polysaccharides from

cells fed GlcNTFA showed a clear UA-GlcNH₂ signal (retention time 7.0 min, 18% abundance, purple trace in the revised Fig. 1C). After treatment with LiOH, the UA-GlcNH₂ signal increased to 74% of the disaccharide (blue trace). We speculate that the formation of UA-GlcNH₂ may be due to hydrolysis of GlcNTFA after extraction, as it underwent a prolonged cleavage reaction (overnight) to ensure sufficient disaccharide cleavage. We have modified Figure 1C and its caption.

Revised Figure 1C:

Revised caption of Figure 1C:

(C) Analysis of disaccharides resulting from LiOH treatment and degradation of bioengineered heparosan (catalyzed by HepIII) using polyamine-based anion exchange (PAMN)-high-performance liquid chromatography (HPLC) with detection at 232 nm. Retention times: Δ UA-GlcNH₂, ~7.0 min; Δ UA-GlcNac, ~25.5 min; Δ UA-GlcNS, ~30.5 min. Orange: Polysaccharides from *E. coli* K5 ASSH fed GlcNac. Red: Polysaccharides from *E. coli* K5 ASSH fed GlcNac; the extracted polysaccharides were treated with LiOH. Purple: Polysaccharides from *E. coli* K5 ASSH fed GlcNTFA. Blue: Polysaccharides from *E. coli* K5 ASSH fed GlcNTFA; the extracted polysaccharides were treated with LiOH. Black: Polysaccharides from *E. coli* K5 ASSH fed GlcNTFA; after treating the extracted polysaccharides with LiOH they were reacted with 3-phosphoadenosine-5-phosphosulfate (PAPS) and NST. Polysaccharides treated with NST and PAPS showed the disappearance of the Δ UA-GlcNH₂ chromatographic peak and the emergence of a Δ UA-GlcNS peak.

2. The enzyme evolution strategies have been largely explored and could be used either alone or in combination. This work employed the PROSS-FRISM platform to enhance the efficiency of NST. However, to generalize this approach, more work and discussion are required. **Firstly, the explanation and schematic representation of this approach was not straightforward enough.** The readers could not easily understand the importance and value of this approach. It seems the authors “rationally” picked several sites based on structural and modeling information and combined them with established PROSS and FRISM approaches. Maybe I did not completely understand the beauty of the semi-rational idea, the PROSS and FRISM approaches are beyond my expertise, but the authors could definitely do better to improve the manuscript to be more readable.

Response: Thank you for your constructive feedback. We have revised Figure 2A to better illustrate the directional evolution of PROSS-FRISM and have modified the Discussion section of the manuscript accordingly. It is important to note that a fully random mutation library entails a substantial workload due to its inherently stochastic nature. Here, in the initial step, we identified 39 key amino acid sites influencing NST stability through structural analysis and molecular dynamics simulations, establishing the framework for subsequent iterative evolution. However, the mutation of all 39 sites proved overly extensive, leading us to use the PROSS framework to directly address 15 sites (resulting in mutant NSTM1). For the remaining 24 sites, we employed the FRISM strategy, which involved Rosetta_ddg-based virtual screening, cluster analysis, and cluster summation to design stable NST mutants to the greatest possible extent (resulting in mutant NSTM8). The PROSS-FRISM strategy showcases a knowledge-driven design process, enhancing the precision and efficiency of the design endeavors.

Revised Figure 2A:

A

Revised Results, line 201:

The library generated by irrational design yields a large and highly random number of mutants. Despite our efforts to optimize screening methods, the process remains labor-intensive. Therefore, we prefer employing rational design methods to narrow down the range of amino acid mutations, resulting in a “small but smart” mutation library. In this study, we used computational and experimental approaches to develop the NST protein with high thermal stability as a template and subsequently designed the PROSS-FRISM evolutionary strategy. Through structural analysis and molecular dynamics simulations, candidate mutation sites were identified based on stability and sequence conservation, thus establishing the subsequent “design scope”(total 39 hot sites). The PROSS engineering approach facilitated the efficient evolution of computationally stable mutations, while the FRISM strategy effectively guided iterative mutations of residual hotspot sites. By combining computational prediction, experimental validation, and iterative optimization, we designed NST mutants with enhanced stability. Therefore, the PROSS-FRISM strategy holds promise as a universal directed evolution tool for eukaryotic proteins expressed in prokaryotic systems.

Since only NST was studied as an example, it is probably challenging to extend to other enzymes. What is the situation of the other modifying enzymes(expression system, stability, activity, yield)? Can the other modifying enzymes like C5-epi, 2-OST, 6-OST, and 3-OST-1 be optimized similarly? Even

though a workflow should at least be described for further optimization of other enzymes. More discussion about the current method for future applications is recommended.

Response: We appreciate your valuable recommendation. Within the heparin synthesis system, NST represents the simplest sulfotransferase, comprising only 325 amino acids. However, due to the characteristics of the sulfotransferase activity and a lack of cell-based high-throughput screening methods for sulfotransferase, there is a need for knowledge-guided strategies to reduce the screening throughput of sulfotransferase. Therefore, NST serves as an excellent example for research and design within the PROSS-FRISM strategy. Subsequent heparin-modifying enzymes, including C5-epi, 2-OST, and 6-OST, are already included in our work plan. Indeed, we have made some progress in the robust protein development of C5-epi and 2-OST using the PROSS-FRISM strategy. Specifically, following the implementation of this design strategy, both the expression levels and stability (measured using 24-h residual enzyme activities) of C5-epi and 2-OST have been improved significantly. The enzyme activity of 2-OST has also been enhanced by >2.2-fold, although there was no significant difference in C5-epi enzyme activity. We speculate that the independent C5-epi forward reaction shows unusual sigmoidal kinetic behavior¹, where C5-epi reactions typically proceed rapidly and reach reaction equilibrium². Research on C5-epi could be focused on enzyme stability development. Detailed reports on the development of robust C5-epimerase, 2-O-sulfotransferase, and other sulfotransferases will be featured in our upcoming publications.

In response to your suggestion, we have added some new discussion points in the manuscript.

Revised Discussion, line 384:

Within the heparin synthesis system, NST represents the simplest sulfotransferase, comprising only 325 amino acids. However, due to the sulfotransferase activity characteristics of the enzyme and the lack of cell-based high-throughput screening methods for sulfotransferase, there is a need for knowledge-guided strategies to reduce the screening throughput of sulfotransferase. Therefore, NST serves as an excellent example for research and design within the PROSS-FRISM strategy. Future research can focus on the semi-rational design of other heparin-modifying enzymes, such as C5-epi, 2-OST, 6-OST, and 3-OST. By employing the PROSS-FRISM strategy, the stability and catalytic efficiency of these enzymes can be enhanced, leading to the development of robust industrial enzyme preparations. The application of the PROSS-FRISM strategy to expand the utility of these recombinant heparin-modifying enzymes will offer greater opportunities in the field of heparin-related research and applications.

3. The chemoenzymatic synthesis of heparin from heparosan has already been reported by several papers before. The author should cite those studies and compare and discuss the strategies. Is it possible to chemically incorporate GlcNS after the harvest of heparosan? Is the process in this study more cost-effective than previously published ones?

Response: Thank you for your valuable insight. In the research content, after obtaining heparin, trimethylamine-trioxide can be used for chemical incorporation of GlcNS. However, issues regarding the environmental friendliness and lack of specificity in chemical synthesis make enzymatic synthesis a more competitive synthetic strategy. In this study, the highly stable mutant NSTM8 maintained >90% enzyme activity for at least 9 days after immobilization, coupled with an eightfold increase in catalytic efficiency compared with that of NST-WT. This suggests a potential reduction of at least 90% in the future application of NST enzyme formulations. Following your suggestions, additional discussion on chemically modifying heparin has been added to the manuscript.

Revised Discussion, line 370:

Chemical N-deacetylation and N-sulfation reactions are typically included in chemical enzymatic heparin synthesis, wherein heparosan is deacetylated using an aqueous solution

of NaOH. Subsequently, the N-sulfation step is conducted using trimethylamine-trioxide sulfur to prepare the intermediate N-sulfated heparosan polysaccharide³. Excessively strong alkaline conditions can cause chain depolymerization reactions, and therefore strict control of the chemical reaction conditions is necessary to regulate the resulting N-sulfated heparin derivatives⁴.

Besides, the chemoenzymatic *de novo* synthesis of heparin oligosaccharides established by Jian Liu's group should be at least discussed in the manuscript. By comparing these approaches, in the author's opinion, what would be **the future direction of production of heparin**? More in-depth discussion was recommended.

Response: We appreciate your valuable guidance. The *de novo* chemoenzymatic approach offers a more efficient and precise method for synthesizing these heparin molecules with defined structures. By *de novo* synthesizing GAG oligosaccharides with precise structures, researchers can better study their interactions with biological molecules and develop novel drugs with improved efficacy and specificity. The *de novo* chemoenzymatic approach for synthesizing oligosaccharide drugs represents a future direction for the heparin industry, including requirements for **non-animal sourced** materials and **homogeneous structures**. Undoubtedly, this entails longer research cycles and processes.

Currently, developing **non-animal sourced** heparin is an urgent goal for the industry, aiming to replace current production lines of unfractionated heparin (UFH) and low-molecular-weight heparin (LMWH) for fundamental anticoagulant needs. This could address concerns related to variability and potential contamination associated with animal-sourced heparin, and pave the way for a more reliable and sustainable supply of this critical anticoagulant. On the basis of your feedback, we have added some new content and discussion on chemoenzymatic *de novo* synthesis.

Revised Introduction, line 50:

To address the deficiency in N-deacetylase activity, the present *in vitro* chemoenzymatic approach employed uridine diphosphate-N-trifluoroacetylglucosamine (UDP-GlcNTFA) instead of the natural substrate of α 1-4-N-acetylglucosaminyltransferase (KifA)^{5,6} or *Pasteurella multocida* heparosan synthase 2 (PmHS2)⁷, uridine diphosphate-N-acetylglucosamine (UDP-GlcNAc). In this method, UDP-GlcNTFA undergoes polymerization

with uridine diphosphate-D-glucuronic acid (UDP-GlcA), and then the trifluoroacetyl group can be efficiently removed by mild alkaline treatment⁸. N-sulfation was achieved by incorporating a *Homo sapiens* N-sulfotransferase (NST) domain with substantial activity. The oligosaccharides containing GlcNS are isomerized at the C5 position⁹, followed by the application of three sulfotransferases (2-OST⁸, 6-OST¹⁰, and 3-OST¹¹) and a sulfonate donor (3'-phosphoadenosine-5'-phosphosulfate, PAPS) to achieve sulfation modification at the O-position of the sugar chain, thus synthesizing structurally uniform anticoagulant heparin oligosaccharides.

Revised Discussion, line 412:

The chemoenzymatic approach is an effective method for preparing structurally defined GAG oligosaccharides that are difficult to synthesize using traditional chemical synthesis methods. This method is suitable for specific biological and pharmacological applications^{12, 13}, representing the future direction of the heparin industry. Non-animal sourced heparin is the primary challenge that the heparin industry must address, to replace current production lines of UFH and low-molecular-weight heparin (LMWH).

4. How was the molecular size (length of sugars) of heparosan controlled? Is it controlled by metabolic engineering or chemical digestion? The MW distribution was not included in the analysis. (Table 2)

Response: Thank you for your valuable comments. We have now included the determination of the molecular weight of polysaccharides produced by *E. coli* K5 ASSH in the same culture conditions at different concentrations of Ac₄GlcNTFA (100 µg/ml, 200 µg/ml, and 500 µg/ml) in Figure S13. The results revealed that **the molecular size is not controlled by the amount of monosaccharide added.**

Meanwhile, the molecular weight of the –GlcNTFA-GlcA– polysaccharide (Fig. S13ABC) is approximately **11 kDa**, while the fully sulfated heparin (K5EH) (Fig. S13D) has a molecular weight of **16 kDa**, consistent with its expected fully sulfated weight. Thus, it is inferred that there is no significant degradation of sugar chains during mild treatment with LiOH. Therefore, **the molecular weight is not regulated by chemical digestion.**

Many existing reports have focused on regulating molecular weight through metabolic engineering¹⁴, but it is not a primary focus of our study. It is worth noting that the molecular

weight of wild-type K5 polysaccharides ranges from **25 to 150 kDa**¹⁵. However, in this study, the *kfiA* gene was knocked out from the K5 strain, and the heparin synthase PmHS2 was introduced, which has lower ability to synthesize higher molecular weight monodisperse polymers than PmHS1¹⁶. Thus, the heparin synthase may be a key factor in regulating molecular weight. Mutation or replacement of the **heparin synthase may directly achieve control over molecular weight**.

In response to your comments, we have revised Figure S13, its caption, and the Results section of the paper.

Revised Figure S13:

Revised caption:

Figure S13. The molecular weight of bioengineered heparin determined by high-performance gel permeation chromatography. (A) Molecular weight determination of engineered K5 polysaccharides from cells cultured using 100 µg/ml Ac4GlcNTFA. (B) Molecular weight determination of engineered K5 polysaccharides from cells cultured using 200 µg/ml Ac4GlcNTFA. (C) Molecular weight determination of engineered K5 polysaccharides from cells cultured using 500 µg/ml Ac4GlcNTFA. (D) Molecular weight determination of K5EH. Note: Mn represents the number average molecular weight, Mp represents the peak position molecular weight, Mw represents the weight average molecular weight, and the unit of molecular weight is Dalton (Da).

Revised Results, line 320:

High-performance gel permeation chromatography was employed to determine the molecular weight of K5EH, which was around 11 kDa before sulfation and 16.5 kDa after full sulfation (Figure S13). This indicates that mild alkaline hydrolysis did not cause significant degradation of the polysaccharide.

Minor comments:

Please carefully re-examine the manuscript for any typos and mistakes. Such as Fig.1A(E.coil for E.coli)

Response: We are thankful for your reminder. The spelling errors in the paper have been corrected.

Responses to the comments of Reviewer #2

Deng et al. present a chemo-enzymatic synthesis strategy for heparin, an important anticoagulant drug. Animal-derived heparin is highly heterogeneous and it has led to substantial health risks in the past. Thus, there is a great ongoing effort in the glycobiology field to synthesize defined heparin molecules. Here, the authors describe a strategy in which they combine bacterial production of heparosan containing a synthetic GlcNAc analog with the in vitro sulfation of the polysaccharide backbone using recombinant enzymes. Two key advancements for carrying out this process are described: first, the use of the unnatural monosaccharide N-trifluoroacetylglucosamine (GlcNTFA) during heparosan production in E. coli K5 cells, which can subsequently be deacetylated chemically allowing to overcome the

limitation of producing a deacetylase enzyme. A second advancement is the engineering of the human N-sulfotransferase (NSF) to increase its thermal stability and activity. The authors use the PROSS-FRISM platform to predict stability increasing mutations, which are then tested by extensive mutational studies of recombinant NST mutant enzymes. The experimental data on increased protein stability and anticoagulation activity looks convincing, however, the presented improvements might not reach the novelty bar needed for this journal.

Response: We are grateful for your insightful feedback. We appreciate the acknowledgment of our work on the chemoenzymatic synthesis strategy for heparin, an essential anticoagulant drug. We would like to reiterate the significance of the innovative research efforts that we have pursued.

In terms of **theoretical innovation**, the efficient recombinant expression of eukaryotic proteins in inexpensive prokaryotic expression systems holds significant importance, yet there is a lack of literature on protein engineering strategies for designing such systems. The PROSS-FRISM strategy proposed in our research adopts a knowledge-guided semi-rational design approach, successfully exploring a promising path between evolutionary cycles. Notably, we conducted screening of approximately 300 mutations in NST in just 6 weeks, ultimately obtaining an enzyme (NSTM8) with 24 robust mutations compared with the wild-type, a mutation rate of 7.7% of the residues in the enzyme. This strategy systematically increases the possibility of introducing beneficial mutations while significantly reducing the experimental workload. We believe this strategy can also be extended to the engineering of other eukaryotic proteins, such as other heparin sulfotransferases.

In terms of **practical innovation**, the production of non-animal sourced heparin represents a primary challenge for the heparin industry. Currently, the biosynthesis of non-animal sourced heparin primarily focuses on recombinant expression and fermentation of NDST¹⁷ or chemical deacetylation of K5 polysaccharide (–GlcNAc-GlcA–)³. However, the optimization of fermentation for NDST still yields limited results, and the chemical deacetylation conditions of K5 polysaccharide (–GlcNAc-GlcA–) are stringent, leading to difficulty in controlling molecular weight degradation. In our study, we incorporated GlcNTFA, commonly used in glycochemistry, into the biosynthesis pathway of K5

polysaccharide, leveraging the tolerance of various enzymes in the synthesis pathway. GlcNTFA can undergo hydrolysis in mild conditions, enabling effective control of sugar chain molecular weight. The synthesized K5EH demonstrates potential efficacy as a substitute for heparin, opening up new avenues for the production of safer and more reliable anticoagulant drugs.

We appreciate your feedback and acknowledge that the significance and value of our study may not have been clearly expressed in the original manuscript. We have made improvements to the manuscript, particularly in the Introduction section, to more clearly articulate the novelty of our work. We will carefully consider the suggestions provided to ensure that our work meets the standards of novelty required for publication in *Nature Communications*. Thank you for your valuable input.

Revised Introduction, line 98:

In this context, leveraging a eukaryotic NST as a paradigm, we endeavor to design PROSS-FRISM methodology for enzyme mutagenesis by seamlessly integrating high-throughput activity screening, computational engineering, and a streamlined codon mutant library. Our overarching goal is to engineer a swift, practical, and resource-efficient solution that not only yields mutant enzymes with superior properties but also pushes the boundaries of experimental efficiency. This innovative approach opens new avenues for exploring rapid upregulation effects, with a laser focus on synergistic interactions among mutations, thereby offering a dependable strategy for “fine-tuning” eukaryotic proteins tailored for recombinant expression in prokaryotic hosts.

In summary, our innovative multicomponent approach involves synthesizing N-sulfated heparin polysaccharides as the foundational material for anticoagulant heparin biosynthesis. We achieve this using metabolically engineered *E. coli* K5 coupled with protein-engineered NST. This methodology avoids the use of NDST, streamlining the production process and lowering production costs. Overall, it enables the direct generation of heparins with potent anticoagulant properties. This optimization of the synthesis process not only enhances economic efficiency but also ensures high-quality and sustainable operations.

Minor comments:

- description of figure 2B concludes that mutant NDST-M1 had highest expression level and

stability, however, figure 2B shows a comparison of activities. How were the expression levels compared?

Response: Owing to the presence of a GST tag in the *N*-terminal segment of NST, the expression levels of NST and its mutants were comparatively assessed by fermenting 1 L of culture medium, followed by determining the loading of NST and each mutant on a GST affinity chromatography column. Protein concentration was measured using the BCA method, with bovine serum albumin used to generate a standard curve.

-What are the yields of purified NST-WT and NST-M8 protein?

Response: The relevant data have been added to Table 1. The yield of NST-WT in 1 L of LB medium was 11.2 ± 0.4 mg/l ($n = 3$), while NST-M8 yielded 50.8 ± 3.38 mg/l ($n = 3$).

- the results section is written lengthily and there are sections that would rather belong to the introduction (description of GlcNTFA strategy) or to the method section. It is hard to understand what are novel results and what is the description of previously established techniques.

Response: Thank you for your suggestion. We have streamlined the summary in the **Discussion** section of the GlcNTFA strategy (first and second paragraphs).

Revised Discussion, line 362:

Heparin is an important anticoagulant drug widely used in medical procedures to prevent the formation of blood clots¹⁸. This study demonstrates the successful synthesis of anticoagulant heparin using a semi-synthetic strategy. This process involves *in vitro* cooperation between the *E. coli* strain K5ASSH and NST mutant NST-M8 for the N-site sulfation of bioengineered heparin polysaccharides¹⁹. The presence of GlcNS residues is crucial for all subsequent modifications in the biosynthesis of heparin sulfate (HS), and this approach effectively manipulates these residues to enable the generation of bioengineered heparin.

In the disaccharide repeat units of the capsule polysaccharides synthesized using our method, a non-natural analogue of N-acetylhexosamine efficiently and site-specifically substitutes in heparin, allowing the reliable and predictable introduction of functional groups onto bacterial surfaces. Chemical N-deacetylation and N-sulfation reactions are typically included in chemoenzymatic heparin synthesis, wherein heparosan is deacetylated using an aqueous solution of NaOH. Subsequently, the N-sulfation step is conducted using trimethylamine-trioxide sulfur to prepare the intermediate N-sulfated heparosan polysaccharide³. Excessively

strong alkaline conditions can cause chain depolymerization reactions, hence strict control of the chemical reaction conditions is necessary to regulate the resulting N-sulfated heparin derivatives⁴. The analogue GlcNTFA can be efficiently incorporated into capsule polysaccharides. The engineered K5 polysaccharide so obtained was subjected to mild enzymatic synthesis for N-sulfation of the polysaccharide, avoiding the need for harsh chemical reaction conditions and the use of difficult-to-recombinantly-express NDST proteins²⁰.

- L46-47 expression of NDST1 has recently also been described in mammalian expression systems (Vallet et al, 2023)

Response: We are grateful for your reminder. Recently, the cryo-electron microscopy structure of NDST1 has been reported (Mycroft-West, *et al.*, 2024), and we have added discussion of NDST to the **Introduction** section.

Revised Introduction, line 46:

Recently, high-resolution cryogenic-electron microscopy (cryo-EM) structures of NDST1 have been obtained^{21,22}; the rich structural information will enhance engineering applications of NDST. However, the expression of active deacetylase proteins remains unattainable in the cost-effective *Escherichia coli* expression system, and even in yeast²³ and insect cell/baculovirus expression systems²⁴.

- L107-108 the authors state "we engineered *E. coli* K5 strain", but I don't see any author from this article in the corresponding reference "29". Other groups in the field should be acknowledged correctly.

Response: We appreciate your reminder. Reference "29" was indeed a citation error, and we have now updated it to the correct citation (which is now reference "33") at that position in the text.

Manuscript:

In previous investigations, we engineered *E. coli* K5 strains capable of producing heparosan polysaccharides with non-natural structures^{32,33}.

Revised Reference:

32. Wang, Y.-J. et al. Imaging of *Escherichia coli* K5 and glycosaminoglycan precursors via targeted metabolic labeling of capsular polysaccharides in bacteria. *Sci. Adv.* 9, eade4770

(2023).

33. Hu, H.-Y. et al. Cell-Based Assay Approaches for Glycosaminoglycan Synthase High-Throughput Screening: Development and Applications. ACS Chem. Biol. 18, 1632–1641 (2023).

Responses to the comments of Reviewer #3

Manuscript by Deng et al introduces an enhanced cell-system based strategy for production of heparin (optimising N-Sulfotransferase activity using a rapid multi-method site directed mutagenesis strategy, PROSS-FRISM, using computational and experimental approaches). Mutagenesis screening resulted in dramatically increased activity of NST (11 fold), combined with improved stability (2.5 fold). When combined with semi-synthesis of substrate via mild weak base treatment of heparosan products containing GlcNAcAZ residues, a bioengineering process to make cell-based anticoagulant heparin was demonstrated, resulting in products with high anti-Xa anticoagulant activity (almost 250 IU/mg). High Xa/IIa activity was also noted, which could be beneficial in clinical use due to lower bleeding risk akin to that achieved by LMWHs. Some additional supportive technical advances are described, including a spectroscopic method for monitoring ST activity via conversion of PAPS sulfate donor pNPS (that allows more HT screening of NST and could be applied to other STS). The methodology appears sound and the data support the conclusions.

Collectively these studies represent a major advance and also demonstrate a strategy for improving enzymes catalytic function for the multi-enzyme cascade require to make highly complex heparin and heparan sulfate products. This work could underpin more robust and efficient future production of novel therapeutics. Some minor revisions are need to render the work suitable for publication.

Response: Thank you for your detailed and positive feedback on our manuscript. We are grateful for your recognition of our enhanced cell-system strategy for heparin production and the advancements made in optimizing N-sulfotransferase activity. The necessary revisions to the manuscript will be made in accordance with your feedback.

Minor comments

1.Fig 1A, misspelling E coli

Response: We appreciate your reminder. The spelling error has been corrected.

2. Fig 2D, why are some mutants designated with a strikethrough ?

Response: We have used underscores to denote excluded mutations for clarity. Figure 2D has been modified accordingly.

Revised Figure 2D:

3. Fig 4D, incorrect symbol for K5EH, should be the brown triangle ??

Response: Thank you for pointing this out. The symbol error has been corrected in Figure 4D.

Revised Figure 4D:

4. Table 2: Is the molecular weight ~ 30,000 correct for UFH. Seems high since for most UFH samples it is around 12-15,000.

Response: Thank you. We have checked, and the molecular weight of the unfractionated

heparin used was 14,000. The relevant content has been modified in Table 2.

5. p11/12: the authors describe the high Xa/IIa anticoagulant activity of engineered K5EH (which is more akin to LMWH properties than UFH) and discuss possible reasons such as the higher sulfation compared to UFH. However, they should also mention that 3-O-sulfation, not measured in these products, could also be a factor, since this is required for anti-Xa but not IIa activity. Whilst not essential for the current publication, if the authors have data on 3-O-sulfation this should be included, or at least described as subject to future studies to characterise the products to underpin possible clinical development.

Response: Thank you for your suggestion. 3-O-Sulfation serves as a crucial determinant for high anti-Xa activity. In accordance with your feedback, we have added discussion of 3-O-sulfation to the Results section.

Revised Results, line 347:

We hypothesize that the presence of more sulfate groups, due to more extensive deacetylation of GlcNTFA, leads to higher anticoagulant activity. Additionally, although disaccharide analysis cannot fully identify disaccharide units containing glucosamine 3-O-sulfation^{25,26}, we have reason to speculate that the high efficiency of the 3OST-1 recombinant enzyme we used results in higher anti-FXa activity, because 3-O-sulfation is essential for anti-FXa anticoagulant activity.

6. Some of the figure panels need to be prepared at higher resolution for better clarity.

Response: Thank you for your valuable feedback. We have replaced the images with higher-resolution images.

References:

1. Vaidyanathan, D. *et al.* Elucidating the unusual reaction kinetics of D-glucuronyl C5-epimerase. *Glycobiology* **30**, 847–858 (2020).
2. Debarnot, C. *et al.* Substrate binding mode and catalytic mechanism of human heparan sulfate d-glucuronyl C5 epimerase. *Proc. Natl. Acad. Sci.* **116**, 6760–6765 (2019).
3. Wang, Z. *et al.* Control of the heparosan N-deacetylation leads to an improved bioengineered heparin. *Appl. Microbiol. Biotechnol.* **91**, 91–99 (2011).
4. Wang, Z. *et al.* Response surface optimization of the heparosan N-deacetylation in

- producing bioengineered heparin. *J. Biotechnol.* **156**, 188–196 (2011).
5. Chen, M., Bridges, A. & Liu, J. Determination of the Substrate Specificities of *N*-Acetyl-D-glucosaminyltransferase †. *Biochemistry* **45**, 12358–12365 (2006).
 6. Deng, J.-Q. *et al.* Heparosan oligosaccharide synthesis using engineered single-function glycosyltransferases. *Catal. Sci. Technol.* **12**, 3793–3803 (2022).
 7. Li, Y. *et al.* Donor substrate promiscuity of the N-acetylglucosaminyltransferase activities of *Pasteurella multocida* heparosan synthase 2 (PmHS2) and *Escherichia coli* K5 KfiA. *Appl. Microbiol. Biotechnol.* **98**, 1127–1134 (2014).
 8. Liu, R. *et al.* Chemoenzymatic Design of Heparan Sulfate Oligosaccharides*. *J. Biol. Chem.* **285**, 34240–34249 (2010).
 9. Sheng, J., Xu, Y., Dulaney, S. B., Huang, X. & Liu, J. Uncovering Biphasic Catalytic Mode of C5-epimerase in Heparan Sulfate Biosynthesis. *J. Biol. Chem.* **287**, 20996–21002 (2012).
 10. Xu, Y. *et al.* Structure Based Substrate Specificity Analysis of Heparan Sulfate 6-O-Sulfotransferases. *ACS Chem. Biol.* **30**.
 11. Moon, A. F. *et al.* Dissecting the substrate recognition of 3-O-sulfotransferase for the biosynthesis of anticoagulant heparin. *Proc. Natl. Acad. Sci.* **109**, 5265–5270 (2012).
 12. Arnold, K. *et al.* Design of anti-inflammatory heparan sulfate to protect against acetaminophen-induced acute liver failure. *Sci. Transl. Med.* **12**, eaav8075 (2020).
 13. Liao, Y.-E. *et al.* Using heparan sulfate octadecasaccharide (18-mer) as a multi-target agent to protect against sepsis. *Proc. Natl. Acad. Sci.* **120**, e2209528120 (2023).
 14. Sheng, L.-L., Cai, Y.-M., Li, Y., Huang, S.-L. & Sheng, J.-Z. Advancements in heparosan production through metabolic engineering and improved fermentation. *Carbohydr. Polym.* **331**, 121881 (2024).
 15. Roy, A. *et al.* Metabolic engineering of non-pathogenic *Escherichia coli* strains for the controlled production of low molecular weight heparosan and size-specific heparosan oligosaccharides. *Biochim. Biophys. Acta BBA - Gen. Subj.* **1865**, 129765 (2021).
 16. Sismey-Ragatz, A. E. *et al.* Chemoenzymatic Synthesis with Distinct *Pasteurella* Heparosan Synthases. *J. Biol. Chem.* **282**, 28321–28327 (2007).
 17. Zhang, Y. *et al.* Synthesis of bioengineered heparin by recombinant yeast *Pichia pastoris*. *Green Chem.* **24**, 3180–3192 (2022).

18. Masuko, S. & Linhardt, R. J. Chemoenzymatic synthesis of the next generation of ultralow MW heparin therapeutics. *Future Med. Chem.* **4**, 289–296 (2012).
19. Zhang, Y. *et al.* Synthesis of bioengineered heparin by recombinant yeast *Pichia pastoris*. *Green Chem.* **24**, 3180–3192 (2022).
20. Sheng, J., Liu, R., Xu, Y. & Liu, J. The Dominating Role of N-Deacetylase/N-Sulfotransferase 1 in Forming Domain Structures in Heparan Sulfate*. *J. Biol. Chem.* **286**, 19768–19776 (2011).
21. Vallet, S. D. *et al.* Functional and structural insights into human N - deacetylase/ N - sulfotransferase activities. *Proteoglycan Res.* **1**, e8 (2023).
22. Mycroft-West, C. J. *et al.* Structural and mechanistic characterization of bifunctional heparan sulfate N-deacetylase-N-sulfotransferase 1. *Nat. Commun.* **15**, 1326 (2024).
23. Zhou, X., Chandarajoti, K., Pham, T. Q., Liu, R. & Liu, J. Expression of heparan sulfate sulfotransferases in *Kluyveromyces lactis* and preparation of 3'-phosphoadenosine-5'-phosphosulfate. *Glycobiology* **21**, 771–780 (2011).
24. Li, Y.-J. *et al.* Characterization of heparan sulfate N-deacetylase/N-sulfotransferase isoform 4 using synthetic oligosaccharide substrates. *Biochim. Biophys. Acta BBA - Gen. Subj.* **1862**, 547–556 (2018).
25. Wang, Z. *et al.* Analysis of 3- O -Sulfated Heparan Sulfate Using Isotopically Labeled Oligosaccharide Calibrants. *Anal. Chem.* **94**, 2950–2957 (2022).
26. Liu, J. & Pedersen, L. C. Emerging chemical and biochemical tools for studying 3- O -sulfated heparan sulfate. *Am. J. Physiol.-Cell Physiol.* **322**, C1166–C1175 (2022).

Reviewers' Comments:

Reviewer #1:

Remarks to the Author:

The author addressed most of my concerns. One more question is that , if the biosynthesis route reported in this study is novel and original? I believe the KfiA's tolerance to UDP-GlcNTFA was already reported. It is not accurate that the authors stated they "first" implemented this metabolic strategy.

Reviewer #2:

Remarks to the Author:

All concerns have been addressed.

Reviewer #3:

Remarks to the Author:

This revised manuscript has addressed my comments and concerns and is suitable for publication.

Response to the reviewer reports on manuscript NCOMMS-23-63535

Dear Editors and Reviewers,

We are deeply appreciative of the constructive and insightful feedback, as it has greatly assisted us in addressing key issues in this work. Enclosed is our detailed response (in **blue**) to all the comments raised by the reviewers. Revisions made to the main manuscript or the supplementary materials are highlighted in **red**.

■ Responses to the comments of Reviewer #1

The author addressed most of my concerns. One more question is that , if the biosynthesis route reported in this study is novel and original? I believe the KfiA's tolerance to UDP-GlcNTFA was already reported. It is not accurate that the authors stated they "first" implemented this metabolic strategy.

Response:We appreciate your suggestion; indeed, the tolerance of the heparosan synthase (including KfiA and PmHS2) towards UDP-GlcNTFA has been reported. Here, we have revised our statement in the abstract by removing the word "first."

Revised Abstract, line 16:

To address this, a metabolic strategy is implemented by introducing the monosaccharide GlcNTFA into engineered *Escherichia coli* K5 to facilitate sulfation modification.